# Preferential Normalizing Flows

**Petrus Mikkola,  Luigi Acerbi**,*  **Arto Klami**\*
Department of Computer Science, University of Helsinki
`first.last@helsinki.fi`

## Abstract

Eliciting a high-dimensional probability distribution from an expert via noisy judgments is notoriously challenging, yet useful for many applications, such as prior elicitation and reward modeling. We introduce a method for eliciting the expert's belief density as a normalizing flow based solely on preferential questions such as comparing or ranking alternatives. This allows eliciting in principle arbitrarily flexible densities, but flow estimation is susceptible to the challenge of collapsing or diverging probability mass that makes it difficult in practice. We tackle this problem by introducing a novel functional prior for the flow, motivated by a decision-theoretic argument, and show empirically that the belief density can be inferred as the function-space maximum a posteriori estimate. We demonstrate our method by eliciting multivariate belief densities of simulated experts, including the prior belief of a general-purpose large language model over a real-world dataset.

## 1 Introduction

Representing beliefs as probability distributions can be useful, particularly as prior probability distributions in Bayesian inference – especially in high-dimensional, non-asymptotic settings where the prior strongly influences the posterior [Gelman et al., 2017] – or as probabilistic alternatives to reward models [Leike et al., 2018, Ouyang et al., 2022]. Our goal is to elicit a complex multivariate probability density from an expert, as a representation of their beliefs. By *expert*, we mean an information source with a belief over a problem of interest, termed *belief density*, which does not permit direct evaluation or sampling. The problem is an instance of expert knowledge elicitation, where the belief is elicited by asking elicitation queries such as quantiles of the distribution [O'Hagan, 2019]. The current elicitation literature (see Mikkola et al. 2023 for a recent overview) focuses almost exclusively on extremely simple distributions, mostly products of univariate distributions of known parametric form. Some isolated works have considered more flexible distributions, for instance quantile-parameterized distributions [Perepolkin et al., 2024] for univariate cases, or Gaussian processes [Oakley and O'Hagan, 2007] and copulas for modelling low-dimensional dependencies [Clemen et al., 2000], but we want to move considerably beyond that and elicit flexible beliefs using modern neural network representations [Bishop and Bishop, 2023]. The main challenges are identifying elicitation queries that are sufficiently informative to infer the belief density while being feasible for the expert to answer reliably, and selecting a model class for the belief density that can represent flexible beliefs without simplifying assumptions but that can still be efficiently estimated.

Normalizing flows are a natural family for representing flexible distributions [Papamakarios et al., 2021]. When using flows for modelling a density $p(\mathbf{x})$, learning is usually based on either a set of samples $\mathbf{x} \sim p(\mathbf{x})$ drawn from the distribution (density estimation; Dinh et al., 2014) or on the log density $\log p(\mathbf{x})$ evaluated at flow samples, $\mathbf{x} \sim q(\mathbf{x})$, in the variational inference formulation [Rezende and Mohamed, 2015]. Neither strategy applies to our setup, since we do not have the luxury of sampling from the belief density and obviously cannot evaluate it either. In addition to

---

*Equal contribution

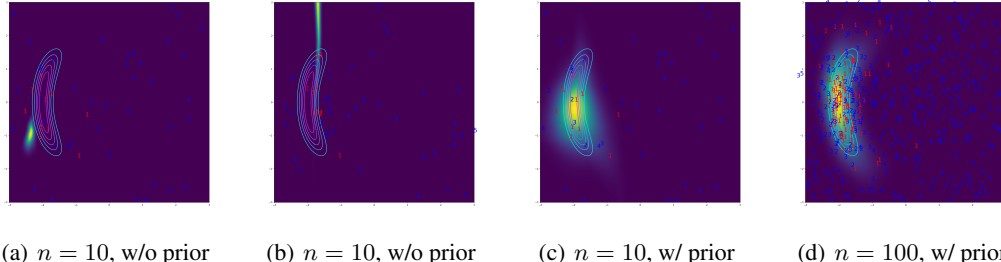

| (a) $n = 10$, w/o prior | (b) $n = 10$, w/o prior | (c) $n = 10$, w/ prior | (d) $n = 100$, w/ prior |

Figure 1: Illustration of belief densities elicited from preferential ranking data by a normalizing flow (contour: true density; heatmap: estimated flow; red: preferred points; blue: non-preferred points). (a)-(b): Typical failure modes of collapsing and diverging mass, when training a flow with just $n = 10$ rankings. (c)-(d): The proposed functional prior resolves the issues, and already with 10 rankings we can learn the correct belief density, matching the result of the flow trained on larger data.

the well-known challenges of training normalizing flows, the setup introduces new difficulties; in particular, a flexible flow easily collapses or finds a way of allocating probability mass in undesirable ways. Significant literature on resolving these issues exists [Behrmann et al., 2021, Salmona et al., 2022, Cornish et al., 2020], but conclusive solutions that guarantee stable learning are still missing. Our solution offers new tools for controlling the flow in low-density areas, and hence we contribute for the general flow literature despite focusing on the specific new task.

We build on established literature on knowledge elicitation for the interaction with the expert. Distributions are primarily characterized by their location and covariance structure, yet humans are notoriously bad at assessing covariances between variables [Jennings et al., 1982, Wilson, 1994]. However, human preferences, with potentially strong interconnections between variables, can be recovered by asking individuals to compare or rank alternatives, a topic studied under discrete choice theory [Train, 2009]. The most studied random utility models (RUMs) interpret human choice as utility maximization with an additive noise component [Marschak, 1959]. To infer the correlation structure in human beliefs indirectly from elicitation data, we study a setup where the expert compares or ranks alternatives (events) based on their probability so that their decisions can be modeled by a RUM. In practice, this means that the data for learning the flow will take the form of *choice sets* $\mathcal{C}_k = \{\mathbf{x}_1, ..., \mathbf{x}_k\}$ of candidates presented to the expert, combined with their choices indicating the preference over the alternatives based on their probability. We stress that candidates $\mathbf{x}$ are here *not* samples from the belief density but are instead provided by some other unknown process, such as an active learning method [Houlsby et al., 2011]. The only information about the belief density comes from the choice.

We are not aware of any previous works that learn flows from preferential comparisons. We first discuss some additional challenges caused by preferential data, and then show how we can leverage preferential structure to improve learning. Specifically, our learning objective corresponds to a function-space maximum a posteriori (FS-MAP), where Bayesian inference is conducted on the function (flow) itself, not its parameters [Wolpert, 1993, Qiu et al., 2024]. The learning objective is exact, in contrast to flow-based algorithms that model phenomena involving discontinuities [Nielsen et al., 2020, Hoogeboom et al., 2021], such as the argmax operator in the RUM model. By construction, the choice sets explicitly include candidates $\mathbf{x}$ that were *not* preferred by the expert, carrying information about relative densities of preferred vs. not preferred points. This allows us to introduce a functional prior that encourages allocating more mass to regions with high probability under a RUM with exponential noise, solving the collapsing and diverging probability mass problem that poses a challenge for flow inference in small data scenarios.

In summary, we introduce the novel problem of inferring probability density from preferential data using normalizing flows and provide a practical solution. We model the expert's choice as a RUM with exponentially distributed noise, and query the expert for comparison or ranking of $k$ alternatives. We derive the likelihoods for $k$-wise comparisons and rankings and study the distribution of the most preferred point among $k$ alternatives, which we term the $k$-wise winner. Based on the interpretation of the $k$-wise winner distribution as a tempered and tilted belief density, we introduce an empirical

function prior and the FS-MAP objective for learning the flow. Finally, we validate our method using both synthetic and real data sets.

## 2   Why learning the density from preferential data is challenging?

Learning flows from small samples is challenging, especially in higher dimensions even when learning from direct data, such as samples from the density. Figure 1 illustrates two common challenges of *collapsing and diverging probability mass*; the illustration is based on our setup to showcase the proposed solution, but the same problems occur in the classical setup. The "collapsing mass" scenario is a form of overfitting, similar to mode collapse in mixture models [Li et al., 2007], but more extreme for flexible models.

In the "diverging mass" problem, the model places probability mass in the regions of low probability. The problem has connections to difficulties in training [Behrmann et al., 2021, Dhaka et al., 2021, Vaitl et al., 2022, Liang et al., 2022] and issues with coupling flows with increasing depth, which tend to produce exponentially large sample values [Behrmann et al., 2021, Andrade, 2024]. One intuitive explanation is that we simply have no information on how the flow should behave far from the training samples, and an arbitrarily flexible model will at least in some cases behave unexpectedly.

If already learning a flow from samples drawn from the density itself is difficult, is it even possible to infer the belief density from preferential data? For instance, for the most popular RUM model (Plackett-Luce; Luce, 1959, Plackett, 1975) we cannot in the noiseless case differentiate between the true density and any normalised positive monotonic transformation of it:

**Proposition 2.1** (Unidentifiability of a noiseless RUM)**.** *Let $p_\star$ be the expert's belief density. For $k \geq 2$, let $\mathcal{D}_{rank} := \{\mathbf{x}_1 \succ \mathbf{x}_2 \succ ... \succ \mathbf{x}_k\}$ be a $k$-wise ranking (see Definition 3.3). If $W \sim Gumbel(0, \beta)$, then for any positive monotonic transformation $g$ holds $\lim_{\beta \to 0} p(\mathcal{D}_{rank}|g \circ p_\star, \beta) = 1$. Proof in B.*

In other words, the *noiseless* solution is not even unique and resolving this requires a way of quantifying the relative utility. Noisy RUM induces such a metric due to the noise magnitude providing a natural scale but even then the belief is identifiable only up to a noise scale; see A for a concrete example for the Thurstone-Mosteller model [Thurstone, 1927, Mosteller, 1951].

Another new challenge is that the candidates $\mathbf{x}$ presented to the expert are given by some external process. In the simplest case, they are drawn independently from some unknown distribution $\lambda(\mathbf{x})$, which does not need to relate to the belief density $p_\star$. We need a formulation that affords estimating $p_\star$ directly, ideally under minimal assumptions on the distribution besides $\lambda(\mathbf{x}) > 0$ for $p_\star(\mathbf{x}) > 0$.

Despite these challenges, we can indeed learn flows as estimates of belief densities as will be explained next, in part by leveraging standard machinery in discrete choice theory to model the expert's choices and in part by introducing a new functional prior for the normalizing flow. The choice process separates the candidate samples $\mathbf{x}$ into preferred and non-preferred ones, and we can use this split to construct a prior that helps learning the flow. That is, the preferential setup also opens new opportunities to address problems in learning flows.

## 3   Random utility model with exponentially distributed noises

The random utility model represents the decision maker's stochastic utility $U$ as the sum of a deterministic utility and a stochastic perturbation [Train, 2009],

$$U(\mathbf{x}) = f(\mathbf{x}) + W(\mathbf{x}), \tag{1}$$

where $f : \mathcal{X} \to \mathbb{R}$ is a deterministic function called *representative utility*, and $W$ is a stochastic noise process, often independent across $\mathbf{x}$. The relationship between these concepts and the task will be made specific in Assumptions 1 to 3. We assume that the domain $\mathcal{X}$ is a compact subset of $\mathbb{R}^d$. Given a set $\mathcal{C} \subset \mathcal{X}$ of possible alternatives, the expert selects a specific opinion $\mathbf{x} \in \mathcal{C}$ through the noisy utility maximization,

$$\mathbf{x} \sim \arg\max_{\mathbf{x}' \in \mathcal{C}} U(\mathbf{x}'). \tag{2}$$

**Definition 3.1** (choice set)**.** Let $k \geq 2$. The *choice set* is a set of $k$ alternatives, denoted by $\mathcal{C}_k = \{\mathbf{x}_1, ..., \mathbf{x}_k\}$. We assume that $\mathcal{C}_k$ is a set of i.i.d. samples from a probability density $\lambda$ over $\mathcal{X}$, but note that the formulation can be generalized to other processes.

For example, if we ask for a pairwise comparison $\mathcal{C}_2 = (\mathbf{x}, \mathbf{x}')$, the expert's answer would be $\mathbf{x} \succ \mathbf{x}'$ if $f(\mathbf{x}) + w(\mathbf{x}) > f(\mathbf{x}') + w(\mathbf{x}')$ for given a realization $w$ of $W$. We denote the random utility model with a representative utility $f$, a stochastic noise process $W$, and a choice set $\mathcal{C}_k$, by $\text{RUM}(\mathcal{C}_k, f, W)$.

We make the common assumption of representing the probabilistic beliefs of a (human) expert in logarithmic form [Dehaene, 2003].

**Assumption 1.** $f(\mathbf{x}) = \log p_\star(\mathbf{x})$; noise is additive for log-density.

**Assumption 2.** $f$ is bounded and continuous.

Inspired by Malmberg and Hössjer [2012, 2014], we assume that the noise is exponentially distributed and thus belongs to the exponential family [Azari et al., 2012].

**Assumption 3.** $W(\mathbf{x}) \sim \text{Exp}(s)$ independently for any $\mathbf{x} \in \mathcal{X}$

With Assumption 1, this corresponds to a model where in the limit of infinitely many alternatives, the expert chooses a point by sampling their belief density (Corollary A.2). The parameter $s$ is here a precision parameter, the reciprocal of the standard deviation of $\text{Exp}(s)$. There are two popular types of preferential queries [Fürnkranz and Hüllermeier, 2011].

**Definition 3.2** ($k$-wise comparison). A preferential query that asks the expert to choose the most preferred alternative from $\mathcal{C}_k$ is called a *k-wise comparison*. The choice is denoted by $\mathbf{x} \succ \mathcal{C}_k$.

**Definition 3.3** ($k$-wise ranking). A preferential query that asks the expert to rank the alternatives in $\mathcal{C}_k$ from the most preferred to the least preferred is a called *k-wise ranking*. The expert's feedback is the ordering $\mathbf{x}_{\pi(\mathcal{C}_k)_1} \succ ... \succ \mathbf{x}_{\pi(\mathcal{C}_k)_k}$ for some permutation $\pi$.

Note that the top-ranked sample of k-wise ranking is the same as the k-wise comparison choice, and the k-wise ranking can be formed as a sequence of k-wise comparisons by repeatedly removing the selected candidate from the choice set, as assumed in the Plackett-Luce model [Plackett, 1975]. Hence, common theoretical tools cover both cases.

## 3.1  The $k$-wise winner

The chosen point of a $k$-wise comparison is central to us for two reasons. First, its distribution provides the likelihood for inference when data come in the format of $k$-wise rankings or comparisons. Second, its distribution in the limit as $k \to \infty$ offers insights for designing a prior that mitigates the challenge of collapsing and diverging probability mass.

**Definition 3.4** ($k$-wise winner). A random vector $X_k^\star$ given by the following generative process is called as *k-wise winner*.

1. Sample $k$-samples from $\lambda(\mathbf{x})$, and denote $\mathcal{C}_k = \{\mathbf{x}_1, ..., \mathbf{x}_k\}$.

2. Sample $\mathbf{x}$ from a Categorical distribution with support $\mathcal{C}_k$ and with probabilities given by $\text{RUM}(\mathcal{C}_k; \log p_\star(\mathbf{x}), \text{Exp}(s))$.

The density of the $k$-wise winner is proportional to the $k$-wise comparison likelihood $p(\mathbf{x} \succ \mathcal{C}_k \mid \mathcal{C}_k)$, namely to $p(\mathbf{x} \succ \mathcal{C}_k \mid \mathcal{C}_k)\lambda(\mathcal{C}_k)$. The likelihood of the $k$-wise comparisons takes the following form.

**Proposition 3.5.** *Let $\mathcal{C}_k$ be a choice set of $k \geq 2$ alternatives. Denote $C = \mathcal{C}_k \setminus \{\mathbf{x}\}$ and $f_C^\star = \max_{\mathbf{x}_j \in C} f(\mathbf{x}_j)$. The winning probability of a point $\mathbf{x} \in \mathcal{C}_k$ equals to*

$$P\left(\mathbf{x} \succ \mathcal{C}_k \mid \mathcal{C}_k\right) = \sum_{l=0}^{k-1} \frac{\exp\left(-s(l+1)\max\{f_C^\star - f(\mathbf{x}), 0\}\right)}{l+1} \sum_{sym:\mathbf{x}_j \in C}^{l} -\exp(-s(f(\mathbf{x}) - f(\mathbf{x}_j))),$$

(3)

*where $\sum_{sym:\mathbf{x}_j \in C}^{l}$ denotes the $l^{th}$ elementary symmetric sum of the set $C$.*

*Proof.* See B. □

The $k$-wise ranking likelihood is a product of the $k$-wise comparison likelihoods where the winners are sequentially removed from the choice set and provided in Appendix A as Equation (A.4).

In the limit of infinitely many comparisons, the $k$-wise distribution reduces to a tempered belief density tilted by the sampling distribution $\lambda$ [Malmberg and Hössjer, 2012, 2014].

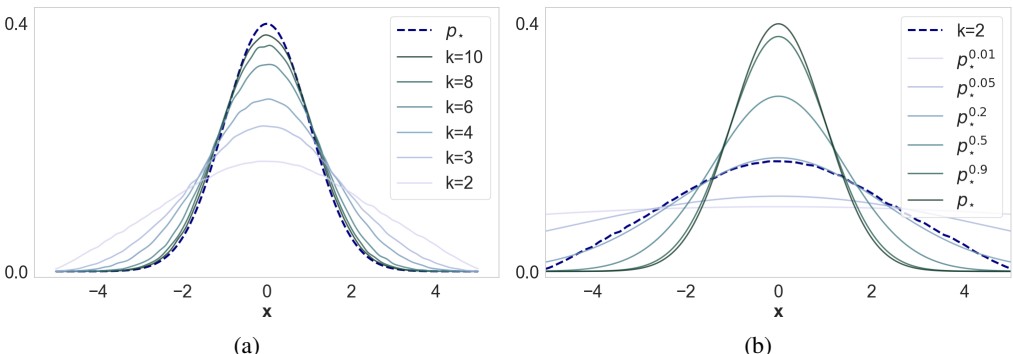

Figure 2: (a) The $k$-wise winner distribution converges to the belief density as $k \to \infty$. (b) The $k$-wise winner distribution can be approximated by a tempered belief density. For example, the tempered belief density with an exponent $1/5$ approximates well the pairwise winner distribution.

**Theorem 3.6.** *If $f$ is bounded and continuous, then the limit distribution of $X_k^\star$ as $k \to \infty$ exists and its density is given by,*

$$p(\mathbf{x}) = \frac{\exp\left(sf(\mathbf{x})\right)\lambda(\mathbf{x})}{\int \exp\left(sf(\mathbf{x}')\right)\lambda(\mathbf{x}')d\mathbf{x}'}. \tag{4}$$

*Proof.* Apply Theorem 18.4 in [Malmberg and Hössjer, 2014] to our setting and note that the first sentence in the proof of Theorem 18.4 is incorrect. For a random variable $Y = X/s$ with $X \sim \text{Exp}(s)$ it holds that $Y \sim \text{Exp}(s^2)$. However, for a random variable $Y = sX$ with $X \sim \text{Exp}(s)$ it holds that $Y \sim \text{Exp}(1)$. Thus, the correct standardization is $Y \leftarrow sY$. $\square$

### 3.2 The $k$-wise winner distribution as a tilted and tempered belief density

Building on the definitions and theorems above, we now introduce the central idea of how to model the belief density based on the $k$-wise winner distribution. The RUM precision parameter $s$ acts as a temperature parameter for the belief density, as $p(\mathbf{x}) \propto \lambda(\mathbf{x})p_\star(\mathbf{x})^s$, by Eq. (4). In general, there is no connection between $\lambda(\mathbf{x})$ and $p_\star(\mathbf{x})$, but intuition can be gained by considering some extreme cases. For $\lambda = p_\star$ we have $p(\mathbf{x}) \propto p_\star(\mathbf{x})^{s+1}$, whereas for uniform $\lambda(\mathbf{x})$ and $s = 1$ the limit distribution is the belief. This is also apparent from Corollary A.2. However, our interest is in cases where $k$ is finite.

For $k < \infty$, forming the $k$-wise winner distribution requires marginalising over the choice set (Proposition 3.5). The formulas can be found in the Appendix (Corollary A.3), and do not have elegant analytic simplifications. However, they empirically resemble tempered versions of the actual belief as illustrated in Figure 2. In other words, finite $k$ plays a similar role as the RUM noise precision $s$. When resorting to $k < \infty$, the choice distribution (Eq. (A.2)) does not match the belief density for the true noise precision $s$, but we can improve the fit by selecting some alternative noise precision such that the choice distribution better approximates the belief. We will later use this connection to build a prior over the flow, and note that for this purpose we do not need an exact theoretical characterization: It is sufficient to know that for some choice of $k$ and $s$ the choice distribution can resemble the target density, at least to the degree that it can be used as a basis for prior information. Given that $k$ is typically fixed, $s$ can be varied in the prior, implying that the further $s$ is from the 'optimal' value, the greater the prior misspecification.

The idea is empirically illustrated in Figure 2. The $k$-wise winner distribution for varying $k$ is shown in Figure 2(a), where the belief density is a truncated standard normal on the interval $[-5, 5]$, comparisons are sampled uniformly over the interval, and $s = 1$. As $k$ increases, the $k$-wise winner distribution approaches the belief density (here $k = 10$ is already very close), but we can equivalently approach the same density by changing the noise level (Figure 2(b)).

# 4 Belief density as normalizing flow

We model the belief density $p_\star$ with a normalizing flow [Rezende and Mohamed, 2015, Papamakarios et al., 2021]. We introduce a new learning principle and objective for the flow which is compatible with any standard flow architecture, as long as it affords easy computation of the flow density. A normalizing flow is an invertible mapping $T$ from a latent space $\mathcal{Z} \subset \mathbb{R}^d$ into a target space $\mathcal{X} \subset \mathbb{R}^d$. $T$ consists of a sequence of invertible transformations, so that the forward (generative) direction $\mathbf{z} \mapsto T(\mathbf{z})$ is fast to compute and the backward (normalizing) direction $\mathbf{x} \mapsto T^{-1}(\mathbf{x})$ is either known in closed form or can be approximated efficiently.

The base distribution on $\mathcal{Z}$ is a simple distribution such as a multivariate normal, whose density is denoted by $p_z$. If we denote the parametrized $T$ by $T_\phi$ given the flow network parameters $\phi$, the parameterized model of the log belief density, denoted by $f_\phi$, can be written as,

$$f_\phi(\mathbf{x}) = \log p_z\left(T_\phi^{-1}(\mathbf{x})\right) + \log|\det J_{T_\phi^{-1}}(\mathbf{x})|, \tag{5}$$

where $J_{T_\phi^{-1}}$ is the Jacobian of $T_\phi^{-1}$. What complicates the learning of $f_\phi$ in our case is the absence of a direct method to sample from $p_\star(\mathbf{x})$, ruling out the conventional algorithms [e.g., Papamakarios et al., 2021]. Instead, we devise a new learning objective explained next.

## 4.1 Function-space Bayesian inference

Our starting point is to perform Bayesian inference for the flow network parameters given preferential dataset $\mathcal{D} = \{(\mathbf{x}^{(i)}, \mathcal{C}_k^{(i)}) \mid \mathbf{x}^{(i)} \succ \mathcal{C}_k^{(i)}\}_{i=1}^n$ ($k$-wise comparisons) or $\mathcal{D} = \{(\sigma^{(i)}, \mathcal{C}_k^{(i)}) \mid \sigma^{(i)}$ is a permutation on $\mathcal{C}_k^{(i)}\}_{i=1}^n$ ($k$-wise rankings),

$$p(\phi \mid \mathcal{D}) \propto p(\mathcal{D} \mid \phi)p(\phi),$$

where $p(\mathcal{D} \mid \phi)$ is the likelihood and $p(\phi)$ is the prior for the flow network parameters. It is difficult to devise a good prior $p(\phi)$, and we instead perform inference over the function [Wolpert, 1993],

$$p(f_\phi \mid \mathcal{D}) \propto p(\mathcal{D} \mid f_\phi)p(f_\phi)$$

where $p(\mathcal{D} \mid f_\phi)$ is the preferential likelihood for comparisons, Eq. (3), or rankings, Eq. (A.4). The function-space prior is easier to specify as we can focus on the characteristics of the log belief density itself, not its parametrization. The function-space prior is often evaluated at a finite set of representer points $\tilde{X} = (\tilde{\mathbf{x}}_1, ..., \tilde{\mathbf{x}}_m)$, where $m$ should be chosen to be large to capture the behavior of the function at high resolution $p(f_\phi(\tilde{X}))$ [Wolpert, 1993, Qiu et al., 2024]. For example, when $f$ is a Gaussian process, the prior representer points in the posterior corresponds to the datapoints (e.g., Equation 3.12 in Rasmussen and Williams, 2006). Following the considerations above, we construct our prior knowledge of $f_\phi$ on a subset of datapoints.

## 4.2 Empirical functional prior

To address the issue of collapsing or diverging probability mass, we introduce an empirical functional prior whose finite marginals at winner points $\{\mathbf{x}_1, ..., \mathbf{x}_n\}$ are independently distributed as

$$p(\mathbf{f}) \propto p_{\text{unif}}(\mathbf{f}) \prod_i \exp(\mathbf{f}_i), \tag{6}$$

where $\mathbf{f} := (f_\phi(\mathbf{x}_1), ..., f_\phi(\mathbf{x}_n))$ and $p_{\text{unif}}$ is an uninformative bounded (hyper) prior that guarantees that the functional prior is proper.

The functional prior Eq. (6) is a special case of a class of priors, $p(\mathbf{f}) \propto \prod_i \boldsymbol{\lambda}_i \exp(s\mathbf{f}_i)$, derived from the following decision-theoretic argument under the exponential RUM model. Let us decompose the preference dataset into winners and losers $\mathcal{D}_k = \mathcal{D}_k^\succ \cup \mathcal{D}_k^{\not\succ}$ by defining $\mathcal{D}_k^\succ := \{\mathbf{x} \mid \exists \mathcal{C}_k \ s.t. \ \mathbf{x} \succ \mathcal{C}_k\}$ and $\mathcal{D}_k^{\not\succ} := \mathcal{D}_k \setminus \mathcal{D}_k^\succ$. The functional prior is the probability of observing only the $k$-wise winners,

$$p(\mathbf{f}) \propto p(\mathcal{D}_k^\succ \mid \mathbf{f}, s, \lambda)p_{\text{unif}}(\mathbf{f}),$$

where $p_{\text{unif}}(\mathbf{f}) \propto 1$ (when $f$ is bounded, Assumption 2). The idea is to consider higher $k$ or $s$ (less noise) than the true ones, as both choices make the density more peaked around the modes of $p_\star$ (see

Figures 2(a) and 2(b)). This choice encourages the flow to place more mass on the winner points in a way that is consistent with the underlying decision model. We consider $k = \infty$ and $s \in (0, \infty)$, where $s$ should be an increasing function of the true $k$. While setting $k = \infty$ reduces the functional prior to a closed form Eq. (4) by Theorem 3.6, the normalizing constant remains difficult. However, for the special case of $\lambda \propto 1$ and $s = 1$, the normalizing constant equals one. We make this choice to retain computational tractability, reminding that the construct is only used as a prior intended for regularizing the solution and does not need to match the true density as such. This comes at the cost of increased prior misspecification, which can, in turn, degrade the quality of the fit, especially when the true value of $k$ is small (compare Figure 4(a) (k=2) versus Figure 1(d) (k=5)).

### 4.2.1 Function-space maximum a posteriori

We train the flow $T_\phi$ by maximizing the unnormalized function-space posterior density of $f_\phi$ conditioned on the preferential data $\mathcal{D} = \mathcal{D}^\succ \cup \mathcal{D}^{\not\succ}$, using stochastic gradient ascent [Kingma and Ba, 2014]. Denoting all points in $\mathcal{D}$ by $\mathbf{X}$ and all winner points in $\mathcal{D}^\succ$ by $\mathbf{X}_\succ$, the training objective is

$$\sum \log \mathcal{L}(\mathcal{D} \mid f_\phi(\mathbf{X}), s) + \sum f_\phi(\mathbf{X}_\succ), \tag{7}$$

where $\mathcal{L}$ is the $k$-wise comparison or ranking likelihood as per Eqs. (3) and (A.4). In the case of ranking data, the winner point $\mathbf{x} \in \mathbf{X}_\succ$ is the first-ranked alternative in each individual $k$-wise ranking, meaning that $\mathbf{x}$ is a $k$-wise winner. A global optimum of Eq. (7) is the function-space maximum a posteriori estimate of the belief density. Pseudo-codes for the overall algorithm (Algorithms 1) and the forward pass for the unnormalized log-posterior (Algorithms 2) are provided in the Appendix. The computational cost of training is similar to standard flow learning from equally many samples.

## 5  Experiments

We first evaluate our method on synthetic data with choices made by simulating the RUM model, to validate the algorithm in cases where the ground truth is known while covering both cases where the responses follow the assumed RUM model and where they do not. We then demonstrate how the method could be used in a realistic elicitation scenario, using a large language model (LLM) as a proxy for a human expert [Choi et al., 2022]. As with a real human, an LLM is unlikely to follow the exact RUM model, but compared to a real user, the LLM expert can tirelessly answer unlimited questions and possible ethical issues and risks relating to human subjects are avoided. LLMs carry their own biases and risks [Tjuatja et al., 2024], but the focus here is on evaluating our algorithm. Code for reproducing all experiments is available at https://github.com/petrus-mikkola/prefflow.

**Setup**  In the main experiments we use $k$-wise ranking with $k = 5$, using relatively few queries to remain relevant for the intended use-cases where the expert's capacity in providing the information is clearly limited. Since learning a preference of higher dimensions is more difficult, we scale the number of queries $n$ linearly with $d$ but still stay substantially below the large-sample scenarios typically considered in flow learning. The details, together with the choice of the flow and the candidate distribution $\lambda$, are provided below for each experiment. As a flow model, we use RealNVP [Dinh et al., 2017] when $d = 2$ and Neural Spline Flow [Durkan et al., 2019] when $d > 2$, implemented on top of [Stimper et al., 2023]. For more details, see Appendix C.4.

**Evaluation**  We assess performance qualiatively via visual comparison of $2d$ and $1d$ marginal distributions between the target belief density and the flow estimate of the belief density, and quantitatively by numerically computing three metrics: the log-likelihood of the preferential data, the Wasserstein distance, and the mean marginal total variation distance (MMTV; Acerbi, 2020) between the target and the estimate. The numerical results are reported as the means and standard deviations of the metrics over replicate runs. As a baseline, we report the results of a method that uses the same preferential comparisons and optimizes the same training objective, but instead of using a flow to represent $\exp(f)$ we directly assume the density is a factorized normal distribution parameterized by means and (log-transformed) standard deviations of all dimensions. This exact method has not been presented in the previous literature, but was designed to validate the merit of the flow representation.

Table 1: Accuracy of the density represented as a flow (*flow*) compared to a factorized normal distribution (*normal*), both learned from preferential data, in three metrics: log-likelihood, Wasserstein distance, and the mean marginal total variation (MMTV). Averages over 20 repetitions (but excluding a few crashed runs), with standard deviations.

| | log-likelihood ($\uparrow$) | | wasserstein ($\downarrow$) | | MMTV ($\downarrow$) | |
|---|---|---|---|---|---|---|
| | *normal* | *flow* | *normal* | *flow* | *normal* | *flow* |
| Onemoon2D | -1.98 ($\pm 0.12$) | -1.09 ($\pm 0.12$) | 0.45 ($\pm 0.04$) | 0.25 ($\pm 0.04$) | 0.30 ($\pm 0.02$) | 0.21 ($\pm 0.02$) |
| Gaussian6D | -1.40 ($\pm 0.07$) | -0.12 ($\pm 0.02$) | 1.74 ($\pm 0.06$) | 1.29 ($\pm 0.05$) | 0.20 ($\pm 0.01$) | 0.09 ($\pm 0.01$) |
| Twogaussians10D | -3.99 ($\pm 0.06$) | -0.09 ($\pm 0.01$) | 7.31 ($\pm 0.12$) | 2.60 ($\pm 0.06$) | 0.47 ($\pm 0.01$) | 0.08 ($\pm 0.00$) |
| Twogaussians20D | -6.35 ($\pm 0.12$) | -0.08 ($\pm 0.01$) | 11.07 ($\pm 0.15$) | 4.55 ($\pm 0.07$) | 0.47 ($\pm 0.00$) | 0.08 ($\pm 0.00$) |
| Funnel10D | -2.21 ($\pm 0.06$) | -0.09 ($\pm 0.01$) | 5.13 ($\pm 0.04$) | 3.92 ($\pm 0.04$) | 0.27 ($\pm 0.00$) | 0.18 ($\pm 0.01$) |
| Abalone7D | -5.53 ($\pm 0.03$) | -2.16 ($\pm 0.12$) | 0.53 ($\pm 0.00$) | 0.34 ($\pm 0.01$) | 0.26 ($\pm 0.00$) | 0.29 ($\pm 0.01$) |

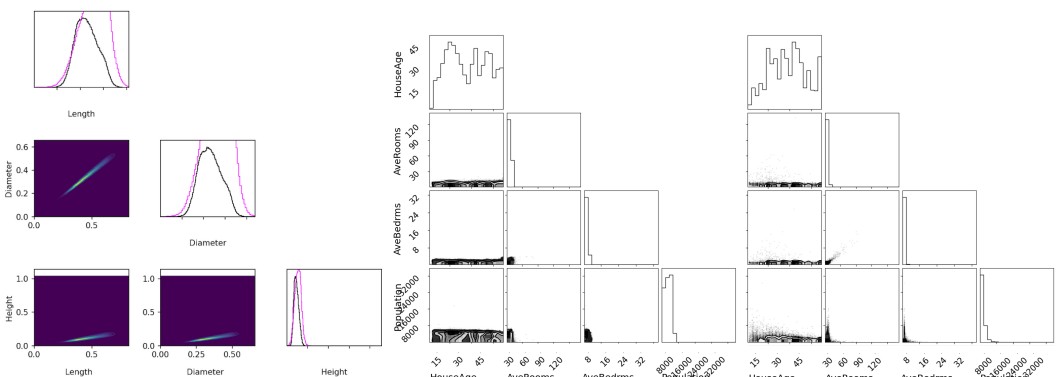

Figure 3: Cross-plot of selected variables of the estimated flow in the Abalone (left) and LLM knowledge elicitation experiment (middle), and the marginal density of the same variables for the ground truth density in the LLM experiment (right). See Figures C.6 and C.7 for other variables.

## 5.1 Synthetic tasks

First, we study the method on synthetic scenarios. For the first set of experiments, we assume a known density $p_\star$ and simulate the preferential responses from the assumed $\mathrm{RUM}(\mathcal{C}_k, \log p_\star, \mathrm{Exp}(1))$. We consider five different target distributions: Onemoon2D, Gaussian6D, Twogaussians10D, Twogaussians20D, and Funnel10D. The densities of the target distributions can be found in Appendix C.1. For all cases we used $100d$ queries and $\lambda$ as a mixture of uniform and Gaussian distribution centered on the mean of the target, with the mixture probability $1/3$ for the Gaussian; this technical simplification ensures a sufficient ratio of the samples to align with the target density even when $d$ is high. Table 1 shows that for all scenarios we can learn a useful flow; all metrics are substantially improved compared to the method based on the normal model and visual inspection (Appendix C.5) confirms the solutions match the true densities well.

**Abalone regression data.** Having validated the method when the queries follow the assumed RUM model with a synthetic belief density, we consider a more realistic target density. We first fit a flow model to the continuous covariates of the regression data *abalone* [Nash et al., 1995], and then use the fitted flow as a ground-truth belief density in the elicitation experiment. The elicitation queries correspond to all $k$-combinations of the dataset size $n = 4177$. The numerical results are again provided in Table 1. Figure 3 shows that the learned flow captures the correlations between variables almost perfectly, which can be hard to see as the flow (heatmap) overlaps the true density (contour). There is some mismatch in the marginals, which is also indicated by the MMTV metric. In terms of the Wasserstein distance and visual comparison (Figure C.8), the flow based method clearly outperforms the baseline.

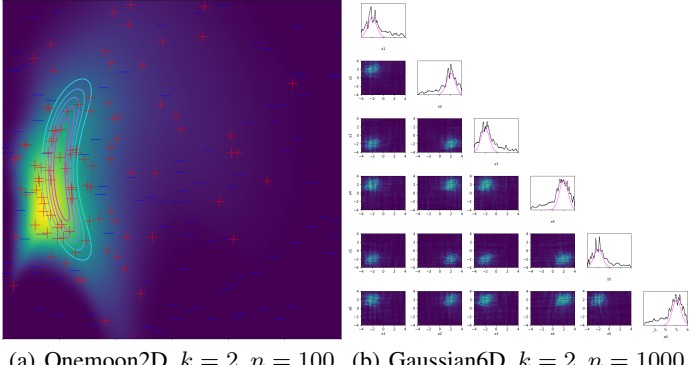

(a) Onemoon2D, $k = 2$, $n = 100$ (b) Gaussian6D, $k = 2$, $n = 1000$

Figure 4: Illustration of belief densities elicited from pairwise comparisons by a normalizing flow.

## 5.2 Expert elicitation with LLM as the expert

In this experiment, we prompt a LLM to provide its belief on how the features of the California housing dataset [Pace and Barry, 1997] are distributed. This resembles a hypothetical expert elicitation scenario, but the human expert is replaced with a LLM (Claude 3 Haiku by Anthropic in March 2024, see Appendix C.2 for the prompt and detailed setup) for easier experimentation. From the perspective of the flow learning algorithm the setup is similar to the intended use-cases.

We query in total 220 $k$-wise rankings through prompting, where the alternatives $\mathcal{C}_k$ are uniformly sampled over the domain specified by 1st and 99th percentiles of each variable in the California housing dataset. The range was chosen to ensure $\lambda(x)$ covers approximately the support of the density, but avoiding outliers. While we lack access to the ground-truth belief density, we can compare the learned LLM's belief density to the empirical data distribution of the California housing dataset, not known for the LLM. Figure 3 shows that there is a remarkable similarity between the distributions such as the marginals of the variables *AveRooms*, *AveBedrms*, *Population*, and *AveOccup* are all correctly distributed on the lower ends of their ranges (which are very broad due to the uniform $\lambda(x)$). Figure D.1 shows that the flow trained without the functional prior of (6) is considerably worse, confirming the FS-MAP estimate is superior to maximum likelihood. While there might be multiple mechanisms for how the LLM forms its knowledge about this specific dataset [Brown et al., 2020], many of the features have clear intuitive meaning. For instance, houses are all but guaranteed to have only a few bedrooms, instead of tens.

## 5.3 Ablation study

We validate the sensitivity of the results with respect to the cardinality of the choice set $k$, the number of comparisons/rankings $n$, the noise level $1/s$, and the choice of distribution $\lambda$ from which the candidates are sampled. In this section, we report a subset of the analysis for varying $k$, while the rest can be found in Appendix D. Table 2 presents the results of experiments on synthetic scenarios (Section 5.1) by varying $k \in \{2, 3, 5, 10\}$ while keeping $n$ fixed. We observe that the accuracy naturally improves as a function of $k$. The common special case in which the expert is queried through pairwise comparisons ($k = 2$) is shown in Figure 4 for the Onemoon2D experiment with $n = 100$ and the Gaussian6D experiment with $n = 1000$. The results indicate that we can already roughly learn the target with $k = 2$ that is most convenient for a user, but naturally with somewhat lower accuracy. For further analysis and more details, see Appendix D. The main takeaway is that low values of $s$ or $k$, especially when $n$ is large, can cause the flow estimate to become overly dispersed due to higher prior misspecification.

## 6 Discussion

Theoretical and empirical analysis validate our main claim: It is possible to learn flexible distributions from preferential data, and the proposed algorithm solves the problem for some non-trivial but

Table 2: Wasserstein distances for varying $k$ across different experiments

|  | $k = 2$ | $k = 3$ | $k = 5$ | $k = 10$ |
|---|---|---|---|---|
| Onemoon2D ($n = 100$) | 0.70 (±0.09) | 0.39 (±0.05) | 0.17 (±0.03) | 0.11 (±0.03) |
| Gaussian6D ($n = 100$) | 2.69 (±0.30) | 2.01 (±0.25) | 1.46 (±0.11) | 1.04 (±0.04) |
| Funnel10D ($n = 500$) | 4.82 (±0.12) | 4.36 (±0.12) | 3.96 (±0.05) | 3.83 (±0.04) |
| Twogaussians10D ($n = 500$) | 5.47 (±0.24) | 3.81 (±0.26) | 2.57 (±0.08) | 2.20 (±0.02) |

synthetic scenarios, with otherwise arbitrary but largely unimodal true beliefs. However, open questions worthy of further investigation remain on the path towards practical elicitation tools.

The method is efficient only for exponential noise with $s = 1$ that gives an analytic prior. Other choices would require explicit normalization or energy-based modeling techniques [Chao et al., 2024]. For a given RUM precision we can, in principle, solve for $k$ such that $s = 1$ becomes approximately correct due to the tempering interpretation (Figure 2), but there are no guarantees that $s = 1$ is good enough for any $k$ sufficiently small for practical use, and this requires an explicit estimate of the noise precision. The ablation studies show that for fixed $k$, increasing $n$ generally improves the accuracy and already fairly small $n$ is sufficient for learning a good estimate (Table D.2). For very large $n$, the accuracy can slightly deteriorate. We believe that this is due to prior misspecification that encourages overestimation of the variation due to the fact that $k$ is finite but in the prior it is assumed to be infinite. Figure D.4 confirms that for a large $n$ the shape of the estimate is extremely close to the target density and the slightly worse Wasserstein distance is due to overestimating the width.

We primarily experimented with k-wise ranging with $k = 5$ and relatively few comparisons. However, we demonstrated that we can learn the beliefs with somewhat limited accuracy already from the most convenient case of pairwise comparisons ($k = 2$), which is important for practical applications. Finally, we focused on the special case of sampling the candidates independently from $\lambda(\mathbf{x})$. In many elicitation scenarios they could be result of some active choice instead, for example an acquisition function maximizing information gain [MacKay, 1992]. The basic learning principle generalizes for this setup, but additional work would be needed for theoretical and empirical analysis.

## 7  Conclusions

The current tools for representing and eliciting multivariate human beliefs are fundamentally limited. This limits the value of knowledge elicitation in general and introduces biases that are difficult to analyze and communicate when the true beliefs do not match the simplified assumed families. Modern flexible distribution models offer a natural solution for representing also complex human beliefs, but until now we have lacked the means of inferring them from ecologically valid human judgements. We provided the first such method by showing how normalizing flows can be estimated from preferential data using a functional prior induced by the setup. Our focus was in specifying the problem setup and validating the computational algorithm, paving way for future applications involving real human judgements. Despite focusing on the scenario where the elicitation judgements are made by a human expert, the algorithm can be used for learning flows from all kinds of comparison and ranking data.

**Broader Impact**  Our goal is to eventually provide methods for accurately characterising human expert beliefs, complementing the existing toolbox with techniques that make less restrictive assumptions and hence support adaptation of better computational tools in a broad range of applications. Knowledge elicitation tools are frequently used e.g. in decision-making and policy recommendations as assistive tools. For such applications, it is critically important to ensure that the mathematical tools are reliable and transparent, and further validation of the methodology is needed before the specific method proposed here could be used in critical applications.

## Acknowledgments and Disclosure of Funding

The work was supported by the Research Council of Finland Flagship programme: Finnish Center for Artificial Intelligence FCAI, and by the grants 345811, 358980, 356498, and 363317. The authors acknowledge support from CSC – IT Center for Science, Finland, for computational resources.

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

# A    Theoretical results

In Section 2 we mentioned that noisy RUM models can be identified for known noise levels. This can be illustrated by the following example:

**Example A.1.** Consider a probit model (Thurstone-Mosteller) model [Thurstone, 1927, Mosteller, 1951]. Denote the probability mass function by $p$, its values at two points $p(\mathbf{x})$ and $p(\mathbf{x}')$, and their difference by $\Delta p = p(\mathbf{x}) - p(\mathbf{x}')$. For a sufficiently large data set of pairwise comparisons, we can estimate the winning probability of $\mathbf{x}$: $P(\mathbf{x} \succ \mathbf{x}') = q$. Since noise follows $N(0, \sigma^2)$, we can deduce that $P(\mathbf{x} \succ \mathbf{x}') = \Phi_{\sigma^2}(\Delta p)$, where $\Phi_{\sigma^2}$ is the cumulative distribution function of $N(0, \sigma^2)$. So, $P(\mathbf{x} \succ \mathbf{x}') = \Phi_{\sigma^2}(\Delta p)$ and $\Delta p = \Phi_{\sigma^2}^{-1}(q)$. Since $p$ is the probability mass function, from a pair of equations we obtain $p(\mathbf{x}) = (1 - \Phi_{\sigma^2}^{-1}(q))/2$ and $p(\mathbf{x}') = (1 + \Phi_{\sigma^2}^{-1}(q))/2$. For the known noise level $\sigma^2$, $p$ is identified.

Section 3 refers to the following corollary that relates the RUM model with sampling.

**Corollary A.2.** *Consider that presenting an $\infty$-wise comparison with the choice set $\mathcal{C} = \mathcal{X}$ to the expert is equivalent to presenting a $k$-wise comparison with large $k$ and points sampled uniformly over $\mathcal{X}$. If the expert choice model follows $RUM(\mathcal{X}; \log p_\star(\mathbf{x}), Exp(1))$, then asking the expert to pick the most likely alternative out of all possible alternatives is equivalent to sampling from their belief density.*

*Proof.* Since $\lambda(\mathbf{x}) = 1/vol(\mathcal{X})$, the terms $\lambda(\mathbf{x})$ and $\lambda(\mathbf{x}')$ in Eq. (4) cancel out. The denominator equals $\int p_\star(\mathbf{x}) d\mathbf{x} = 1$, because $f(\mathbf{x}) = \log p_\star(\mathbf{x})$ and $s = 1$. Thus, $p(\mathbf{x} \succ \mathcal{X}) = \exp(sf(\mathbf{x})) = p_\star(\mathbf{x})$. $\qquad\square$

**Corollary A.3.** *For $k = 2$, the probability density of $X_k^\star$ equals to*

$$p_{X_k^\star}(\mathbf{x}) = 2\lambda(\mathbf{x}) \int_\mathcal{X} F_{Laplace(0,1/s)} \left(\log p_\star(\mathbf{x}) - \log p_\star(\mathbf{x}')\right) \lambda(\mathbf{x}') d\mathbf{x}' \tag{A.1}$$

*For $2 < k < \infty$, the probability density of $X_k^\star$ is proportional to*

$$p_{X_k^\star}(\mathbf{x}) \propto \lambda(\mathbf{x}) \int_{\mathcal{X}^{k-1}} P\left(\mathbf{x} \succ \mathcal{C}_k \mid \mathcal{C}_k\right) d\lambda(\mathcal{C}_k \setminus \{\mathbf{x}\}), \tag{A.2}$$

*where $P\left(\mathbf{x} \succ \mathcal{C}_k \mid \mathcal{C}_k\right)$ is given by Proposition 3.5.*
*For $k = \infty$, the probability density of $X_k^\star$ equals to*

$$p_{X_k^\star}(\mathbf{x}) = C\lambda(\mathbf{x})p_\star^s(\mathbf{x}), \tag{A.3}$$

*where $C > 0$. If $\lambda(\mathbf{x}) = 1/vol(\mathcal{X})$ and $s = 1$, then $C\lambda(\mathbf{x}) = 1$.*

*Proof.* Case $k = 2$.

$$p_{X_k^\star}(\mathbf{x}) \propto \int_\mathcal{X} P(\mathbf{x} \succ \mathbf{x}' \mid p_\star, s)\lambda(\mathbf{x}')\lambda(\mathbf{x})d\mathbf{x}'$$

The normalizing constant can be computed by using Fubini's theorem. Since $P(\mathbf{x} \succ \mathbf{x}' \mid p_\star)\lambda(\mathbf{x}')\lambda(\mathbf{x})$ is $\mathcal{X} \times \mathcal{X}$ integrable, it holds that

$$\int_\mathcal{X} \int_\mathcal{X} P(\mathbf{x} \succ \mathbf{x}' \mid p_\star, s)\lambda(\mathbf{x}')\lambda(\mathbf{x})d\mathbf{x}'d\mathbf{x} = \int_{\mathcal{X} \times \mathcal{X}} P(\mathbf{x} \succ \mathbf{x}' \mid p_\star, s)\lambda(\mathbf{x}')\lambda(\mathbf{x})d(\mathbf{x}', \mathbf{x}) = 0.5,$$

by the symmetry and the fact that $P(\mathbf{x} \succ \mathbf{x}' \mid p_\star, s) + P(\mathbf{x}' \succ \mathbf{x} \mid p_\star, s) = 1$. So, the normalizing constant is 2. By Proposition 3.5 and after straightforward algebraic manipulations, $P\left(\mathbf{x} \succ \mathcal{C}_k \mid \mathcal{C}_k\right)$ simplifies to $F_{Laplace(0,1/s)}\left(\log p_\star(\mathbf{x}) - \log p_\star(\mathbf{x}')\right)$, where $F_{Laplace(0,1/s)}$ is the cumulative distribution function of the Laplace distribution with a location parameter of 0 and a scale parameter of $1/s$.

Case $k = \infty$. It follows from Theorem 3.6. $\qquad\square$

$k$**-wise ranking likelihood**

The $k$-wise ranking likelihood $P\left(\mathbf{x}_{\pi(\mathcal{C}_k)_1} \succ ... \succ \mathbf{x}_{\pi(\mathcal{C}_k)_k} \mid \mathcal{C}_k\right)$ can be computed as a product of $k$-wise comparison likelihoods,

$$\prod_{j=1}^{k-1} P\left(\mathbf{x}_{\pi(\mathcal{C}_k)_j} \succ \mathcal{C}^{(j)} \mid \mathcal{C}^{(j)}\right), \tag{A.4}$$

where $\mathcal{C}^{(1)} = \mathcal{C}_k$, ..., $\mathcal{C}^{(k-1)} = \mathcal{C}_k \setminus \{\mathbf{x}_{\pi(\mathcal{C}_k)_1}, ..., \mathbf{x}_{\pi(\mathcal{C}_k)_{k-2}}\}$.

# B  Proofs

This section provides the proofs for the propositions made in the main paper.

**Proposition.** *Let $p^*$ be the expert's belief density. Denote $N = k - 1$, so that $\mathcal{D}_N = \{\mathbf{x}_1 \succ \mathbf{x}_2 \succ ... \succ \mathbf{x}_N \succ \mathbf{x}_{N+1}\}$. If $W \sim Gumbel(0, \beta)$, then for any positive monotonic transformation $g$, and for $f \equiv g \circ p^*$ it holds,*

$$p(\mathcal{D}_N | f) \xrightarrow{\beta \to 0} 1.$$

*Proof.* Let $f(\mathbf{x}) = g(p^*(\mathbf{x}))$. Then, by Yellott Jr [1977],

$$p(\mathcal{D}_N | f, \beta) = \prod_{n=1}^{N+1} \frac{e^{\frac{1}{\beta} f(\mathbf{x}_n)}}{\sum_{i=n}^{N+1} e^{\frac{1}{\beta} f(\mathbf{x}_i)}} = \prod_{n=1}^{N+1} \frac{e^{\frac{1}{\beta} g(p^*(\mathbf{x}_n))}}{\sum_{i=n}^{N+1} e^{\frac{1}{\beta} g(p^*(\mathbf{x}_i))}}. \tag{B.1}$$

By the product law for limits,

$$\begin{aligned}
\lim_{\beta \to 0+} p(\mathcal{D}_N | f, \beta) &= \prod_{n=1}^{N+1} \lim_{\beta \to 0+} \frac{e^{\frac{1}{\beta} g(p^*(\mathbf{x}_n))}}{\sum_{i=n}^{N+1} e^{\frac{1}{\beta} g(p^*(\mathbf{x}_i))}} \\
&= \prod_{n=1}^{N+1} \mathbb{I}\left(g(p^*(\mathbf{x}_n)) = \max_{n \leq i \leq N+1} g(p^*(\mathbf{x}_i))\right) \\
&= \prod_{n=1}^{N+1} \mathbb{I}\left(p^*(\mathbf{x}_n) = \max_{n \leq i \leq N+1} p^*(\mathbf{x}_i)\right) \\
&= \prod_{n=1}^{N+1} \mathbb{I}\left(p^*(\mathbf{x}_n) = p^*(\mathbf{x}_n)\right) \\
&= 1.
\end{aligned}$$

The second equation holds because the softmax converges pointwise to the argmax in the limit of the temperature approaches zero. The third equation holds because $g$ preserves the order. The fourth equation holds because $p^*(\mathbf{x}_1) > p^*(\mathbf{x}_2) > ... > p^*(\mathbf{x}_{N+1})$. $\qquad\square$

**Proposition.** *Let $\mathcal{C}_k$ be a choice set of $k \geq 2$ alternatives. Denote $C = \mathcal{C}_k \setminus \{\mathbf{x}\}$ and $f_C^\star = \max_{\mathbf{x}_j \in C} f(\mathbf{x}_j)$. The winning probability of a point $\mathbf{x} \in \mathcal{C}_k$ equals to*

$$P\left(\mathbf{x} \succ \mathcal{C}_k \mid \mathcal{C}_k\right) = \sum_{l=0}^{k-1} \frac{\exp\left(-s(l+1)\max\{f_C^\star - f(\mathbf{x}), 0\}\right)}{l+1} \sum_{sym:\mathbf{x}_j \in C}^{l} -\exp(-s(f(\mathbf{x}) - f(\mathbf{x}_j))),$$

*where $\sum_{sym:\mathbf{x}_j \in \mathcal{C}_k \setminus \{\mathbf{x}\}}^{l}$ denotes the $l^{th}$ elementary symmetric sum of the set $C$.*

*Proof.* Fix $\mathbf{x} \in \mathcal{C}_k$, and for any $w \geq 0$ denote $\mathbf{1}_w = \mathbb{I}_{\{f(\mathbf{x})+w \geq f_{\mathcal{C}}^\star\}}$.

$$
\mathrm{P}\left(\mathbf{x} \succ \mathcal{C}_k \mid \mathcal{C}_k\right) = \mathrm{P}\left(\bigcap_{x_j \in C} \{f(\mathbf{x}) + W(\mathbf{x}) > f(\mathbf{x}_j) + W(\mathbf{x}_j)\}\right)
$$

$$
= \int \mathrm{P}\left(\bigcap_{x_j \in C} \{f(\mathbf{x}) + W(\mathbf{x}) > f(\mathbf{x}_j) + W(\mathbf{x}_j)\} \mid W(\mathbf{x})\right) \mathrm{P}(dW(\mathbf{x}))
$$

$$
= \int_0^\infty \mathrm{P}\left(\bigcap_{x_j \in C} \{W(\mathbf{x}_j) < f(\mathbf{x}) + w - f(\mathbf{x}_j)\} \mid w\right) s e^{-sw} dw
$$

$$
= \int_0^\infty s e^{-sw} \prod_{x_j \in C} \mathrm{P}\left(\{W(\mathbf{x}_j) < f(\mathbf{x}) + w - f(\mathbf{x}_j)\} \mid w\right) dw
$$

$$
= \int_0^\infty s e^{-sw} \prod_{x_j \in C} \left(1 - e^{-s(f(\mathbf{x})+w-f(\mathbf{x}_j))}\right) \mathbb{I}_{\{f(\mathbf{x})+w \geq f(\mathbf{x}_j)\}} dw
$$

$$
= \int_0^\infty s e^{-sw} \prod_{x_j \in C} \frac{1}{e^{sw}} \left(e^{sw} - e^{-s(f(\mathbf{x})-f(\mathbf{x}_j))}\right) \mathbf{1}_w dw
$$

$$
= \int_0^\infty s e^{-ksw} \mathbf{1}_w \prod_{x_j \in C} \left(e^{sw} - e^{-s(f(\mathbf{x})-f(\mathbf{x}_j))}\right) dw
$$

Denote $c_j := -\exp(-s(f(\mathbf{x}) - f(\mathbf{x}_j)))$. Let $b_l$ be the $l^{th}$ elementary symmetric sum of the $c_j$ over $j$s. The $l^{th}$ elementary symmetric sum is the sum of all products of $l$ distinct $c_j$ over $j$s. We can write,

$$
\int_0^\infty s e^{-ksw} \mathbf{1}_w \prod_{x_j \in C} \left(e^{sw} - e^{-s(f(\mathbf{x})-f(\mathbf{x}_j))}\right) dw
$$

$$
= \int_0^\infty s e^{-ksw} \mathbf{1}_w \sum_{l=0}^{k-1} b_l e^{(k-1-l)sw} dw
$$

$$
= s \int_0^\infty \sum_{l=0}^{k-1} b_l e^{(k-1-l)sw - ksw} \mathbf{1}_w dw
$$

$$
= s \sum_{l=0}^{k-1} \frac{b_l}{s(l+1)} \int_0^\infty s(l+1) e^{-s(l+1)w} \mathbf{1}_w dw
$$

$$
= \sum_{l=0}^{k-1} \frac{b_l}{l+1} \int_{\max\{f_{\mathcal{C}}^\star - f(\mathbf{x}), 0\}}^\infty s(l+1) e^{-s(l+1)w} dw
$$

$$
= \sum_{l=0}^{k-1} \frac{b_l(1 - G_{s(l+1)}(\max\{f_{\mathcal{C}}^\star - f(\mathbf{x}), 0\}))}{l+1},
$$

$$
= \sum_{l=0}^{k-1} \frac{b_l \exp\left(-s(l+1)\max\{f_{\mathcal{C}}^\star - f(\mathbf{x}), 0\}\right)}{l+1},
$$

with convention that $b_0 = 1$ and $G_\eta$ denotes the cumulative distribution function of $\mathrm{Exp}(\eta)$. $\qquad \square$

| **Algorithm 1** Full algorithm | **Algorithm 2** FS-Posterior$(\phi|\mathcal{D})$ |
|---|---|
| **require:** preferential data $\mathcal{D}_{\text{full}}$ | **require:** precision $s$ |
| **while** not converged **do** | **input:** flow parameters $\phi$, mini-batch $\mathcal{D}$ |
| $\quad$ sample mini-batch $\mathcal{D} \sim \mathcal{D}_{\text{full}}$ | $\mathbf{X}$ = design matrix of $\mathcal{D}$ |
| $\quad \Delta\phi \propto \nabla_\phi \text{FS-Posterior}(\phi|\mathcal{D})$ | $\mathbf{X}_\succ$ = winner points of $\mathbf{X}$ |
| **end while** | loglik = $\sum \log \mathcal{L}(\mathcal{D} \mid f_\phi(\mathbf{X}), s)$ |
| | logprior = $\sum f_\phi(\mathbf{X}_\succ)$ |
| | **return:** loglik + logprior |

## C Experimental details

### C.1 Target distributions

The logarithmic unnormalized densities of the target distributions used in the synthetic experiments are listed below.

$$\textbf{Onemoon2D}: \quad -\frac{1}{2}\left(\frac{\|\mathbf{x}\| - 2}{0.2}\right)^2 - \frac{1}{2}\left(\frac{x_0 + 2}{0.3}\right)^2$$

$$\textbf{Gaussian6D}: \quad -\frac{1}{2}(\mathbf{x} - \boldsymbol{\mu})^T\Sigma^{-1}(\mathbf{x} - \boldsymbol{\mu}), \quad \boldsymbol{\mu} = 2\begin{pmatrix}(-1)^1 \\ (-1)^2 \\ \vdots \\ (-1)^6\end{pmatrix}, \ \Sigma = \begin{pmatrix}\frac{6}{10} & 0.4 & \cdots & 0.4 \\ 0.4 & \frac{6}{10} & \cdots & 0.4 \\ \vdots & \vdots & \ddots & \vdots \\ 0.4 & 0.4 & \cdots & \frac{6}{10}\end{pmatrix}$$

$$\textbf{Twogaussians}: \quad \log\left(\frac{1}{2}\exp\left(\log\mathcal{N}(\mathbf{x} \mid \boldsymbol{\mu}, \Sigma_1)\right) + \frac{1}{2}\exp\left(\log\mathcal{N}(\mathbf{x} \mid \boldsymbol{\mu}, \Sigma_2)\right)\right),$$

$$\sigma^2 = 1, \ \rho = 0.9, \ d \in \{10, 20\}, \ \boldsymbol{\mu} = 3\mathbf{1}_d, \ \Sigma_1 = \begin{bmatrix}\sigma^2 & \rho\sigma^2 & \rho\sigma^2 & \cdots & \rho\sigma^2 \\ \rho\sigma^2 & \sigma^2 & \rho\sigma^2 & \cdots & \rho\sigma^2 \\ \rho\sigma^2 & \rho\sigma^2 & \sigma^2 & \cdots & \rho\sigma^2 \\ \vdots & \vdots & \vdots & \ddots & \vdots \\ \rho\sigma^2 & \rho\sigma^2 & \rho\sigma^2 & \cdots & \sigma^2\end{bmatrix},$$

$$\Sigma_2 = \begin{bmatrix}\sigma^2 & -\rho\sigma^2 & \rho\sigma^2 & \cdots & (-1)^{d-1}\rho\sigma^2 \\ -\rho\sigma^2 & \sigma^2 & -\rho\sigma^2 & \cdots & (-1)^{d-2}\rho\sigma^2 \\ \rho\sigma^2 & -\rho\sigma^2 & \sigma^2 & \cdots & (-1)^{d-3}\rho\sigma^2 \\ \vdots & \vdots & \vdots & \ddots & \vdots \\ (-1)^{d-1}\rho\sigma^2 & (-1)^{d-2}\rho\sigma^2 & (-1)^{d-3}\rho\sigma^2 & \cdots & \sigma^2\end{bmatrix}$$

$$\textbf{Funnel10D}: \quad -\left(\frac{(x_0 - 1)^2}{a^2}\right) - \sum_{i=1}^{10-1}\left(\log(2\pi\exp(2bx_0)) + \frac{(x_i - 1)^2}{\exp(2bx_0)}\right), \ a = 3, b = 0.25$$

### C.2 LLM expert elicitation experiment

The version of the Claude 3 model used in the LLM expert elicitation experiment was *claude-3-haiku-20240307*. In the experiment, we used the following prompt template to query the LLM. The configurations specified within the XML tags <configuration> were sampled from $\lambda$, a uniform distribution. The prompt template is as follows:

```
Data definition:
<data>
California Housing
```

We collected information on the variables using all the block groups
in California from the 1990 Census. In this sample a block group on
average includes 1425.5 individuals living in a geographically
compact area. Naturally, the geographical area included varies
inversely with the population density. We computed distances among
the centroids of each block group as measured in latitude and longitude.
This dataset was derived from the 1990 U.S. census, using one row
per census block group. A block group is the smallest geographical
unit for which the U.S. Census Bureau publishes sample data
(a block group typically has a population of 600 to 3,000 people).
A household is a group of people residing within a home.

Number of Variables: 8 continuous
Variable Information:
- MedInc median income (expressed in hundreds of thousands of dollars)
in block group
- HouseAge median house age in block group
- AveRooms average number of rooms per household
- AveBedrms average number of bedrooms per household
- Population block group population
- AveOccup average number of household members
- Latitude block group latitude
- Longitude block group longitude
</data>

The variables are:
<variables>
{MedInc, HouseAge, AveRooms, AveBedrms, Population, AveOccup, Latitude,
Longitude}
</variables>
always reported in this order.

Given these combinations of variables below, please list them from
most likely to least likely in your opinion. Consider what each
variable represents and its realistic value in light of the properties
of the dataset.

<configurations>
A=0.79,0.81,0.40,0.60,0.74,0.49,0.59,0.75,0.00,0.04
B=0.09,0.10,0.22,0.92,0.16,0.95,0.02,0.91,0.25,0.02
C=0.72,0.50,0.17,0.70,0.37,0.78,0.15,0.14,0.05,0.05
D=0.39,0.69,0.27,0.63,0.25,0.13,0.81,0.89,0.31,0.02
E=0.34,0.52,0.01,0.34,0.90,0.42,0.49,0.02,0.26,0.04
</configurations>

<task>
1. First, think your answer step by step, considering the model
and data definition in detail.
2. Then discuss each combination separately in light of your
thoughts about data. Do not assign an ordering yet.
3. Finally, summarize all your considerations.
4. At the end, write your final ordering as a comma-separated
list of letters within an XML tag <order></order>.
</task>

Table C.1: Accuracy of the density represented as a flow (*flow*) compared to a factorized normal distribution (*normal*), both learned from preferential data, in three metrics: log-likelihood, Wasserstein distance, and the mean marginal total variation (MMTV). Averages over 20 repetitions (but excluding a few crashed runs), with standard deviations.

| | log-likelihood ($\uparrow$) | | wasserstein ($\downarrow$) | | MMTV ($\downarrow$) | |
|---|---|---|---|---|---|---|
| | *normal* | *flow* | *normal* | *flow* | *normal* | *flow* |
| Abalone7D | -5.53 ($\pm 0.03$) | -2.16 ($\pm 0.12$) | 0.53 ($\pm 0.00$) | 0.34 ($\pm 0.01$) | 0.26 ($\pm 0.00$) | 0.29 ($\pm 0.01$) |
| mod-Abalone7D | -5.25 ($\pm 0.07$) | -3.52 ($\pm 0.09$) | 1.05 ($\pm 0.00$) | 0.65 ($\pm 0.01$) | - | - |

## C.3 Modified Abalone7D experiment

By modifying Abalone7D experiment, we can consider a synthetic technical validation constructed so that the data distribution is more realistic. We do this by mis-using a regression data set so that the response variable is interpreted as indication of preference and the queries are formed by presenting the expert a choice between different samples. If we denote by $g(\mathbf{x}_i)$ the regression function for the $i$th covariate set, then $\mathbf{x}_i$ is chosen over $\mathbf{x}_j$ if $g(\mathbf{x}_i) > g(\mathbf{x}_j)$. We remark that the task itself is not particularly interesting as the response variable does not correspond to any real belief (instead, we learn a distribution over the covariates for which the response variable is high), but it is still useful for validating the algorithm as we now need to cope with choice sets that do not match any simple distribution $\lambda(\mathbf{x})$. Instead, the choice sets are now formed by uniform sampling over the sample indices, which means they are drawn from the marginal distribution of the covariates. Note that this is different from the target density, which is the density of covariates for samples with high response variables.

In the experiment, we do not assume any noise on the expert response. Hence, the expert follows a noiseless RUM with the representative utility $g(\mathbf{x}_i)$ equals to the response variable of $i$th covariate $\mathbf{x}_i$. This means that the choice distribution resembles Dirac delta function at the points with the highest response variables. For this reason, we cannot compute the MMTV metric as it involves integrating over the absolute differences of the marginals, which leads to numerical issues. The numerical results are provided in Table C.1 and the visual results in Figure C.1.

## C.4 Other experimental details

**Hyperparameters**. In all the experiments, we use the value $s = 1$ in the preferential likelihood regardless of how misspecified it is with respect to the ground-truth model. Neural Spline Models have 2 hidden layers and 128 hidden units. The number of flows is 6, 8, or 10 depending on the problem complexity. RealNVP models have 4 hidden layers and 2 hidden units. The number of flows is 36 when the number of rankings is more than 50, and 8 otherwise. Other architecture-specific details correspond to the default values implemented in the *normflows* package, a PyTorch-based implementation of normalizing flows [Stimper et al., 2023].

**Optimization details**. Models are trained for a varying number of iterations from $10^5$ to $5 \times 10^5$ with the Adamax optimizer [Kingma and Ba, 2014] and varying batch size from 2 to 8. The learning rate varies from $10^{-5}$ to $5 \times 10^{-5}$ depending on the problem dimensionality, with higher learning rates for higher-dimensional problems. A small weight decay of $10^{-6}$ was applied.

**Computational environment**. Models are trained and evaluated on a server with nodes of two Intel Xeon processors, code name Cascade Lake, with 20 cores each running at 2.1 GHz. Double precision numbers were used to improve the stability of the training process. We did not explicitly record the training times or memory consumption, but note that the considered data sets and flow architectures are all relatively small.

**Experiment replications**. Every experiment was replicated with 20 different seeds, ranging from 1 to 20, but a few replications failed due to not well-known reasons, sometimes due to memory issues and sometimes due to numerical instabilities that led the replication to crash. The results are reported over the completed runs. In the main experiment table (Table 1), there was one failed replication in the Twogaussians20D experiment and two in the Onemoon2D experiment.

**Dataset licence**: *Abalones*: (CC BY 4.0) license, original source [Nash et al., 1995]

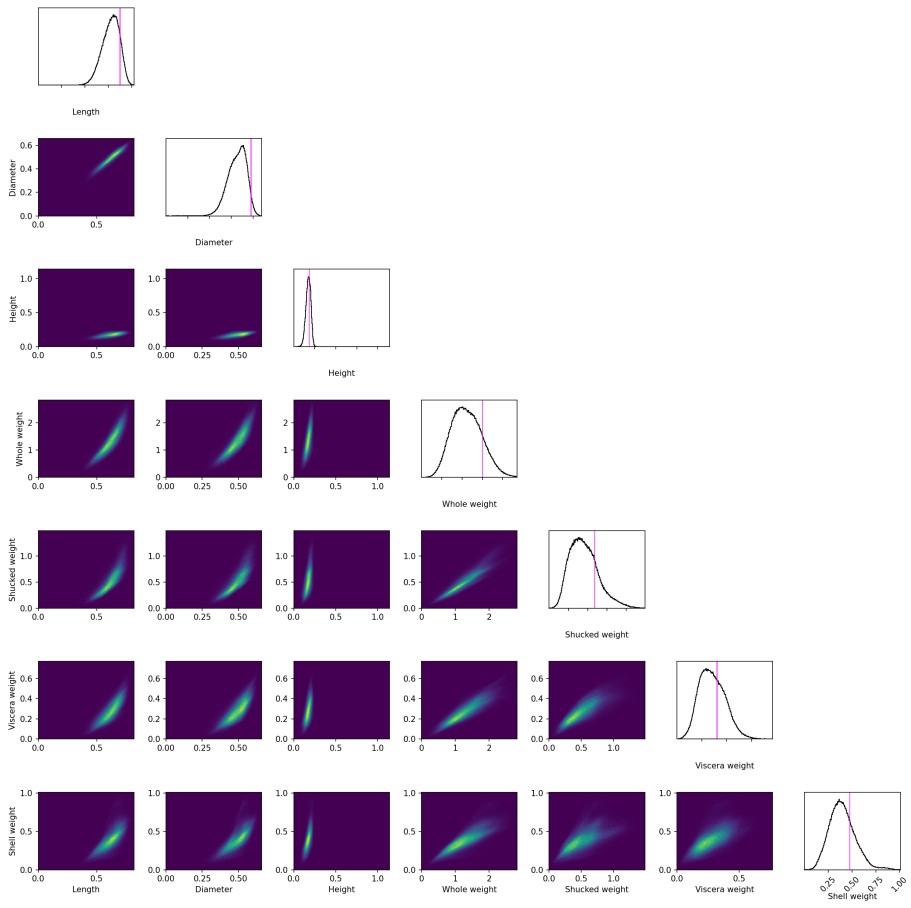

Figure C.1: Full result plot for the modified Abalone7D experiment, where the target (unnormalized) belief density corresponds to the abalone age.

## C.5 Plots of learned belief densities

Figure 3 presented a subset of the cross-plots for the multivariate densities for the Abalone regression data and the LLM experiment. Here, we provide the complete visual illustrations over the full density for both, as well as corresponding visualisations for all of the other experiments in Figures C.2 to C.8.

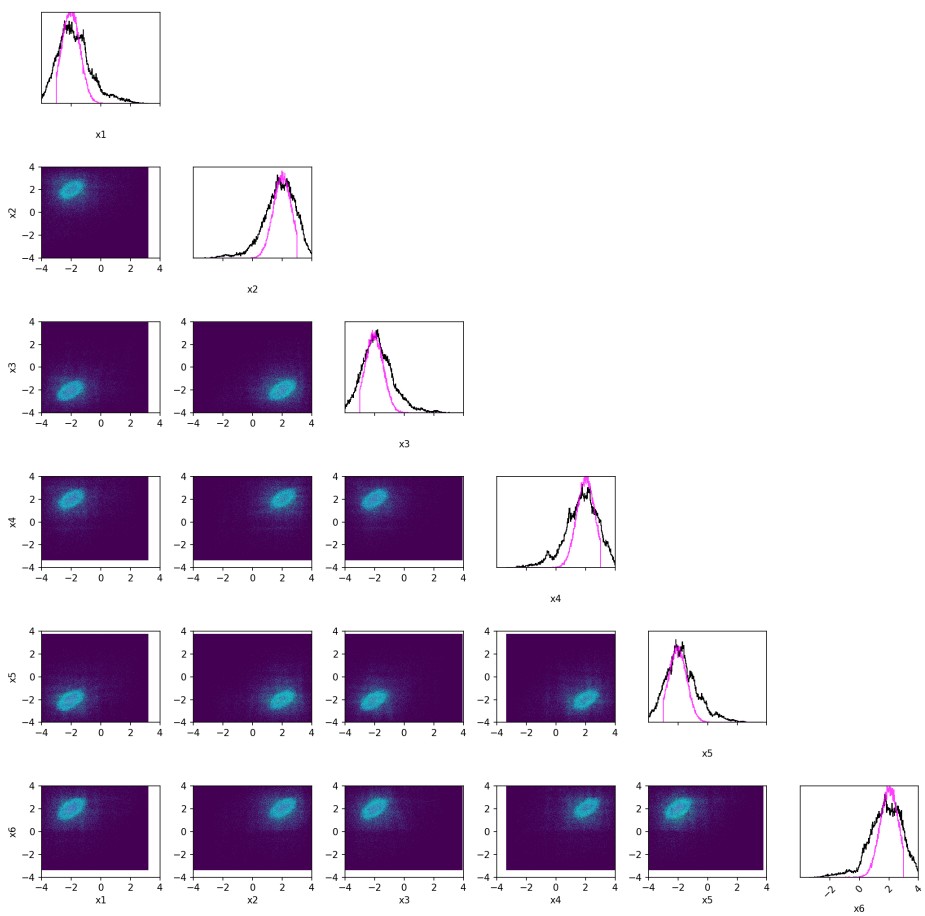

Figure C.2: Gaussian6D experiment. The target distribution is depicted by light blue contour points and its marginal by a pink curve. The learned flow is depicted by dark blue contour sample points and its marginal by a black curve.

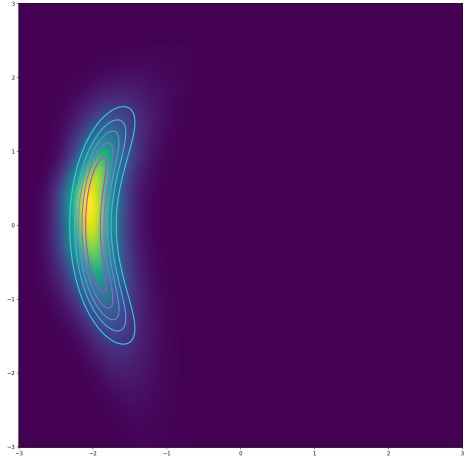

Figure C.3: Estimated belief density for the Onemoon2D data. See Figure 1 for other visualisations on the same density.

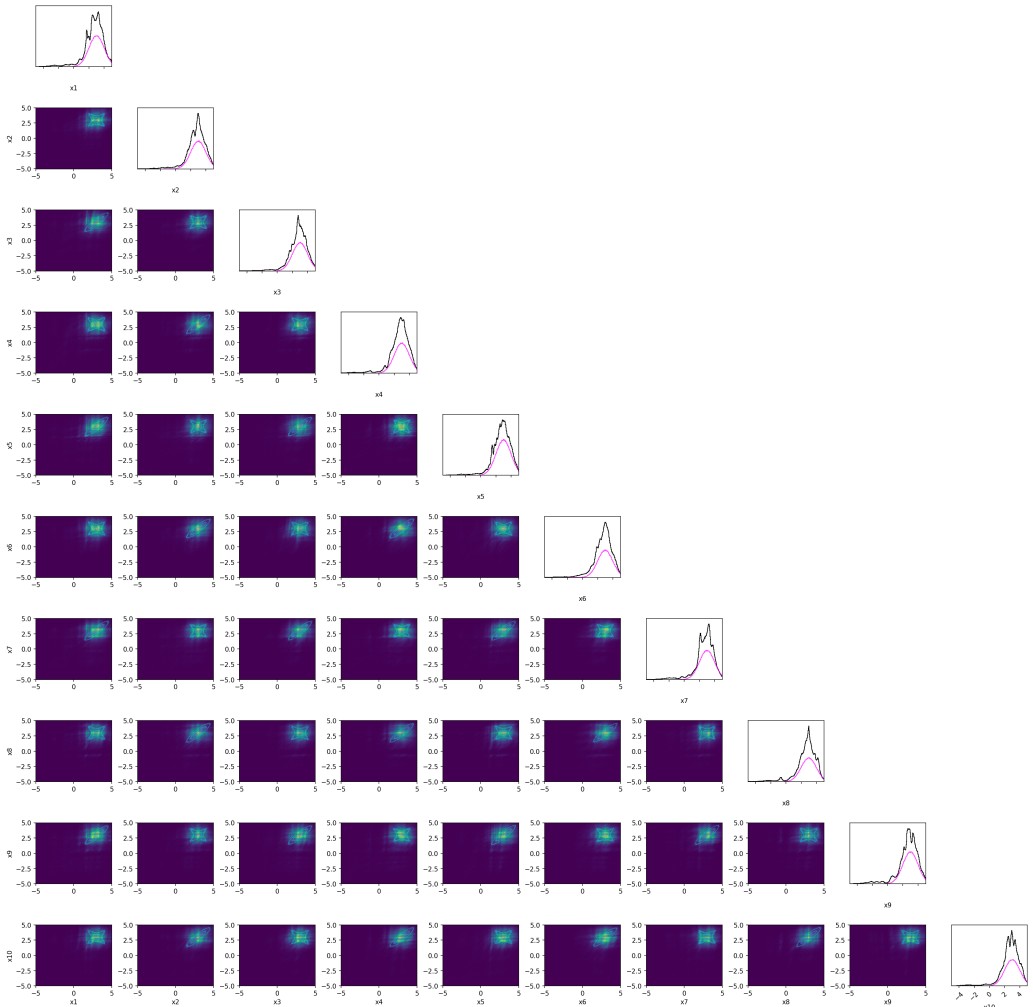

Figure C.4: Twogaussians10D experiment. The target distribution is depicted by light blue contour points and its marginal by a pink curve. The learned flow is depicted by dark blue contour sample points and its marginal by a black curve.

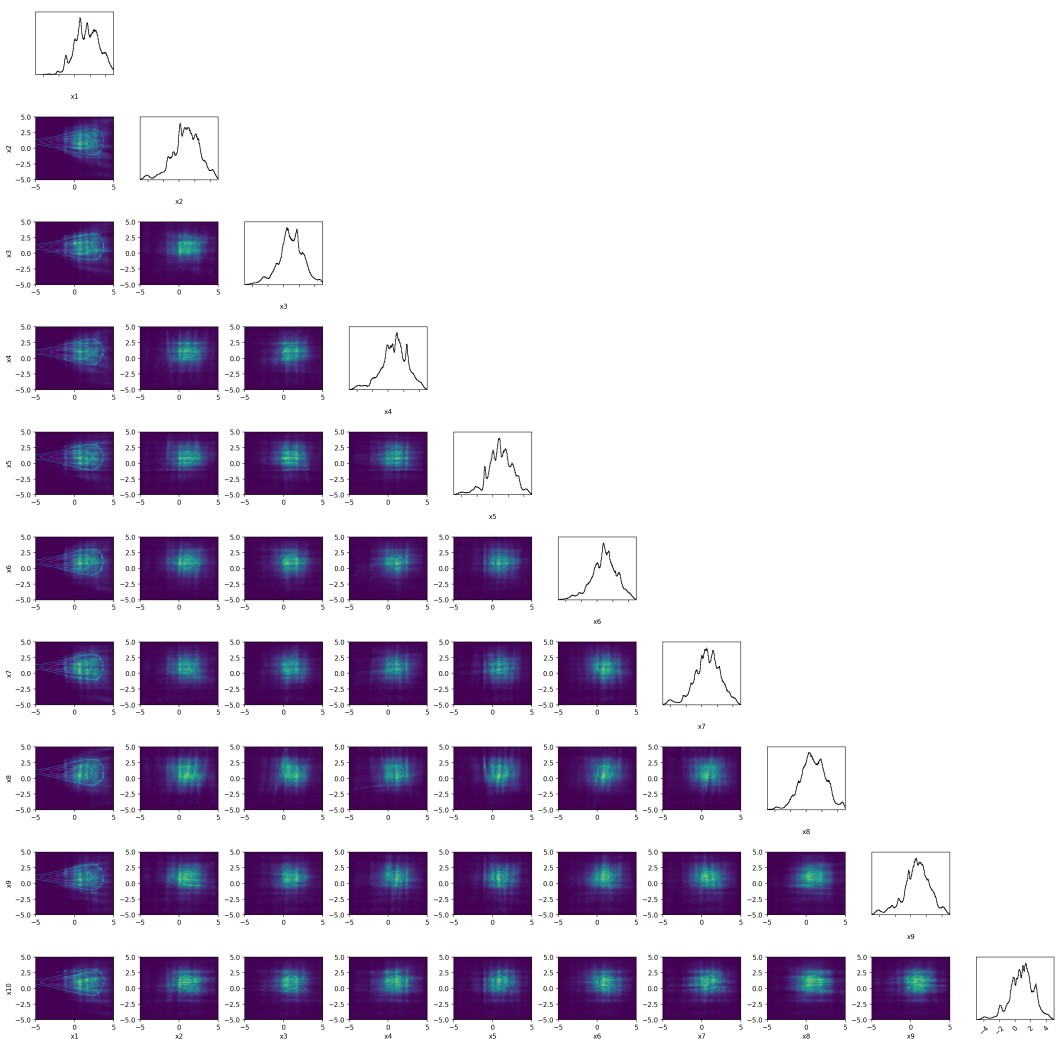

Figure C.5: Estimated belief density for the Funnel10D data. The narrow funnel dimension ($x1$) is extremely difficult to capture accurately, but the flow still extends more in that dimension, seen as clear skew in all marginal histograms.

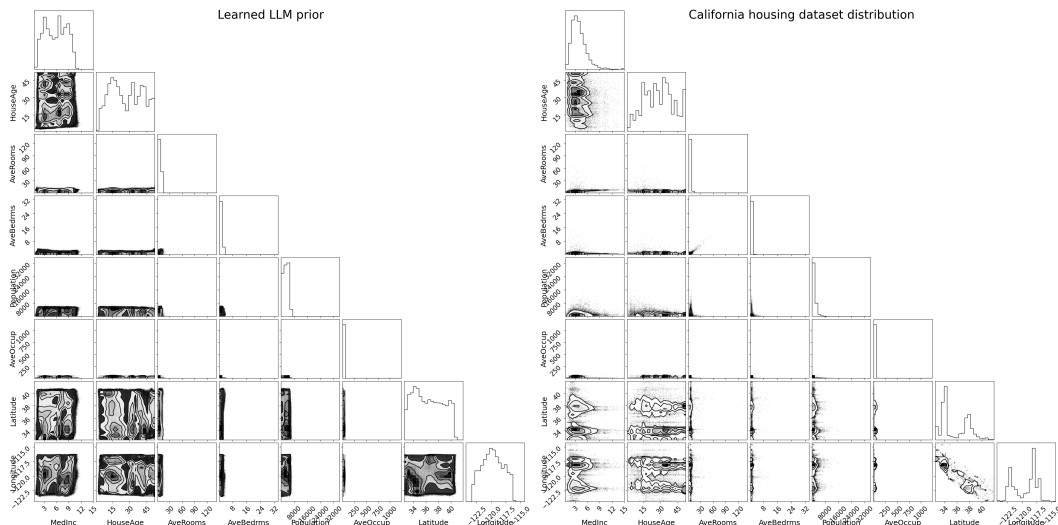

Figure C.6: Full result plot for the LLM expert elicitation experiment, complementing the partial plot presented in Figure 3.

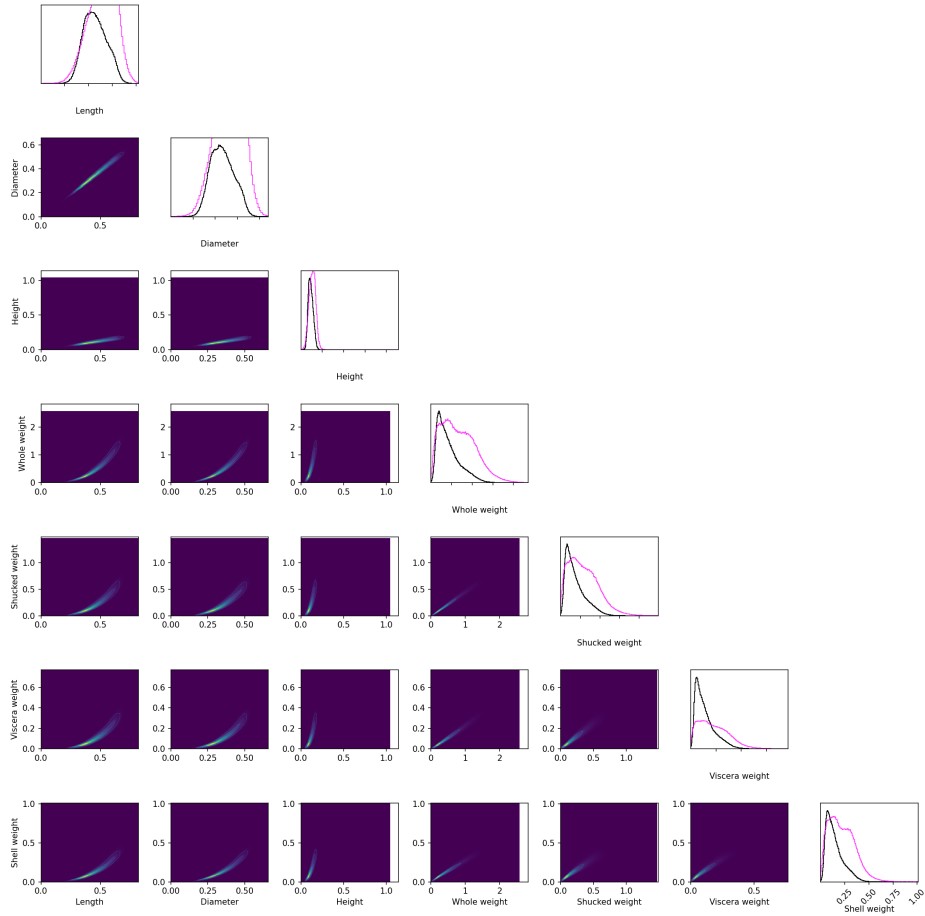

Figure C.7: Full result plot for the Abalone7D experiment, complementing the partial plot presented in Figure 3. The target distribution is depicted by light blue contour points and its marginal by a pink curve. The learned flow is depicted by dark blue contour sample points and its marginal by a black curve.

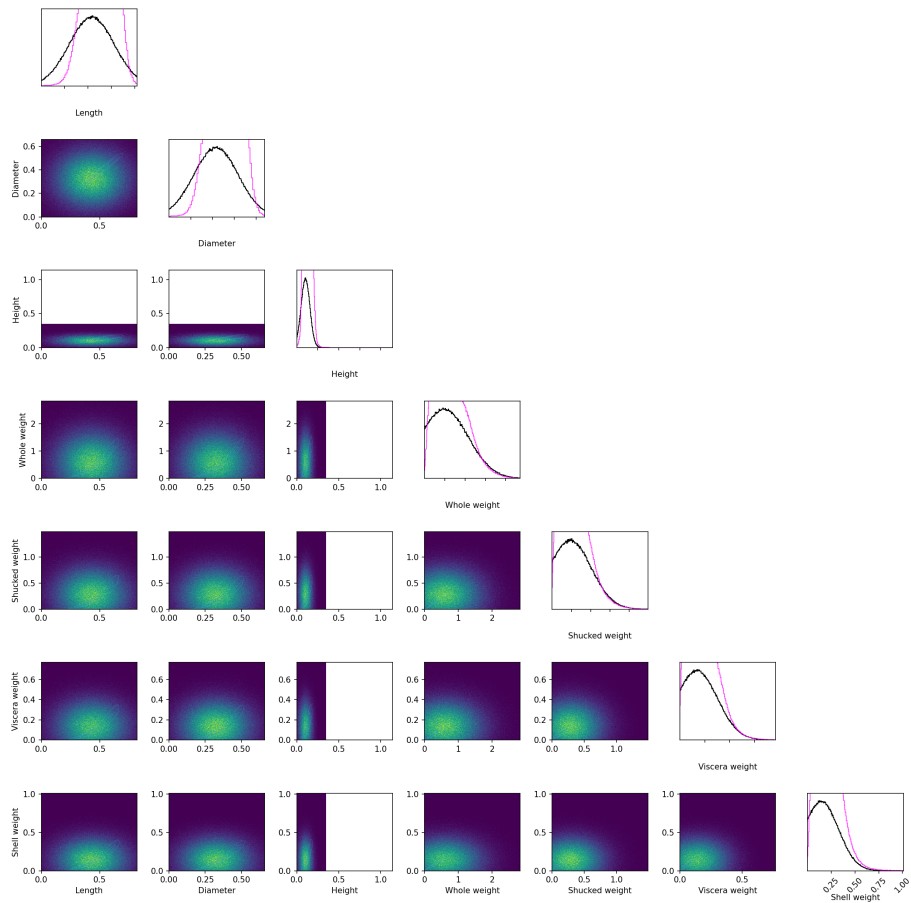

Figure C.8: Full result plot for the Abalone7D experiment for the baseline method.

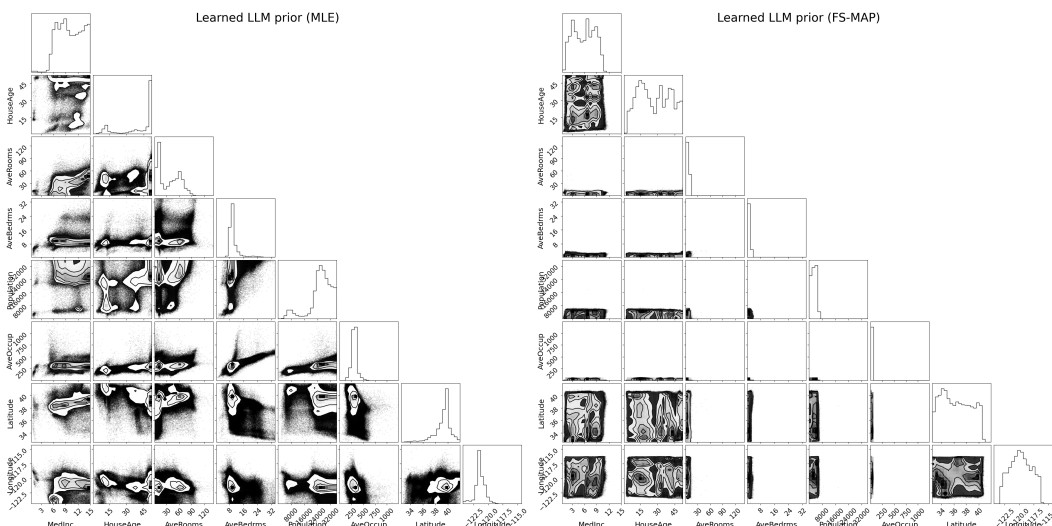

Figure D.1: The effect of the functional prior on the learned belief density in Experiment 5.2. The left plot corresponds to learning the LLM's belief density using maximum likelihood in the training, and the right plot to using function-space maximum a posteriori with the proposed functional prior. We hypothesize that the extreme marginals (e.g. median income) obtained from maximum likelihood estimation are due to problems with collapsing or diverging probability mass.

Table D.1: The means of the variables based on (first row) the distribution of California housing dataset and the sample from the preferential flow fitted to the LLM's feedback trained (second row) with the likelihood only and (third row) with the both likelihood and prior.

|  | MedInc | HouseAge | AveRooms | AveBedrms | Population | AveOccup | Lat | Long |
|---|---|---|---|---|---|---|---|---|
| True data | 3.87 | 28.64 | 5.43 | 1.1 | 1425.48 | 3.07 | 35.63 | -119.57 |
| Flow w/ prior | 9.83 | 43.01 | 125.18 | 8.07 | 22983.76 | 1290.0 | 28.81 | -117.94 |
| Flow w/o prior | 5.91 | 27.19 | 6.28 | 1.58 | 2868.52 | 3.37 | 36.43 | -119.75 |

# D    Ablation studies

This section reports additional experimentation to complement the results presented in the main paper. Unless otherwise stated, the rest of the details in the experiments are as discussed in Sections 5 and C.4. The only exception is the number of flows, which are scaled by the number of rankings $n$ to increase flexibility in line with the available data. However, when $n$ is as in the main paper, the number of flows remains unchanged.

## D.1    Effect of the functional prior

Figure D.1 shows the effect of the functional prior for the LLM experiment, showcasing how the maximum likelihood estimate learning the flow without the functional prior exhibits the diverging mass property. Table D.1 summarizes the densities in a quantitative manner by reporting the means for all variables. The table shows how the solution without the prior can be massively off already in terms of the mean estimate, for instance having the mean number of rooms at 125.

## D.2    Effect of the noise level $1/s$

Figure D.2 investigates the interplay of the true RUM noise and the assumed noise in the preferential likelihood on the Onemoon2D data.

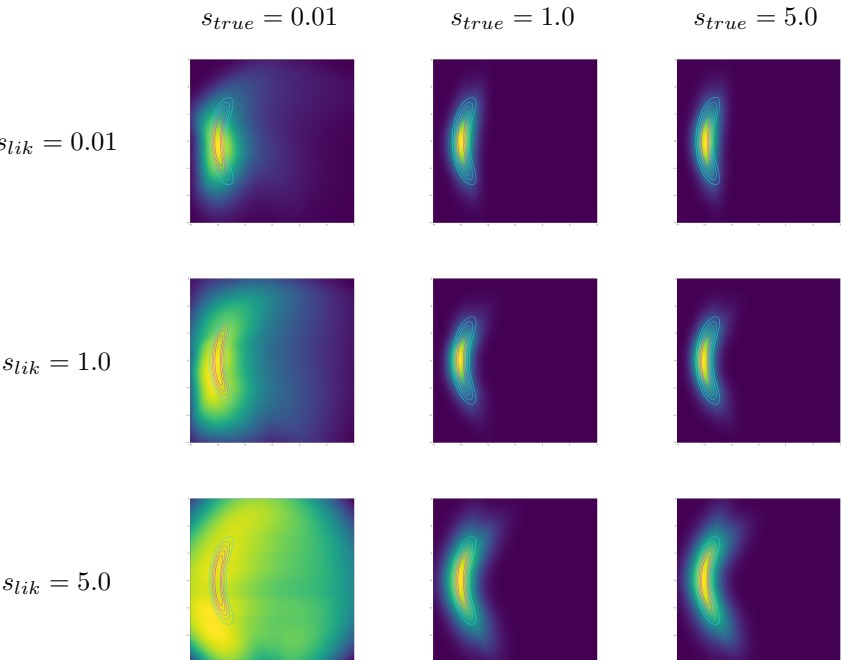

Figure D.2: Preferential flow fitted via FS-MAP with varying precision levels in the data generation process (in RUM) $s_{true}$, and precision levels in the preferential likelihood $s_{lik}$. The first column shows that a lower precision level in RUM leads to a more spread fitted flow, as expected. The middle plot is the only scenario where both the likelihood and the functional prior are correctly specified, resulting in the best result. Since the prior is misspecified in the bottom-right plot, the best results are not achieved, contrary to expectations. However, this misspecification does not lead to catastrophic performance deterioration but rather to a more spread-out fitted flow.

### D.3 Effect of the candidate sampling distribution $\lambda$

We validate the sensitivity of the results in terms the choice of the distribution $\lambda$ that the candidates are sampled from. Figure D.3 studies the effect of $\lambda$, the unknown distribution from which the candidates to be compared are sampled from, complementing the experiment reported in Section 5.1 and confirming the method is robust for the choice. In the original experiment the candidates were sampled from a mixture distribution of uniform and Gaussian distribution centered on the mean of the target, with the mixture probability $w = 1/3$ for the Gaussian. Figure D.3 reports the accuracy as a function of the choice of $w$ for one of the data sets (Onemoon2D), so that $\lambda$ goes from uniform to a Gaussian, and includes also an additional reference point where $\lambda$ equals the target. For all $w > 0.5$ we reach effectively the same accuracy as when sampling from the target itself, and even for the hardest case of uniform $\lambda$ (with $w = 0$ the distance is significantly smaller than the reference scale comparing the base distribution with the target.

### D.4 Effect of the number of rankings $n$

We validate the sensitivity of the results in terms of the number of comparisons/rankings $n$. Table D.2, as well as Figures D.4 and D.5, report the results of an experiment that studies the effect of $n$.

Increasing $n$ generally improves the accuracy and already fairly small $n$ is sufficient for learning a good estimate (Table D.2). For very large $n$, the accuracy can slightly deteriorate. We believe that this is due to prior misspecification that encourages overestimation of the variation due to the fact that $k$ is finite but in the prior it is assumed to be infinite. In the Onemoon2D experiment, Figure D.4 confirms that for $n = 1000$ the shape of the estimate is extremely close and the slightly worse Wasserstein distance is due to overestimating the width. The same holds for other experiments such as Twogaussians10D illustrated in Figure D.5.

Figure D.3: The Onemoon2D experiment replicated for varying sampling distributions $\lambda$ from where the candidates are sampled. In original Onemoon2D, $\lambda$ is a mixture of uniform and Gaussian distribution centered on the mean of the target, with the mixture probability $w = 1/3$ for the Gaussian. Here, $\lambda \in \{\text{Uniform}, \text{Gaussian-Uniform Mixture}, \text{Target}\}$ with letting the mixture probability $w$ to vary in $\{0.1, 1/4, 1/3, 1/2, 2/3, 3/4, 1.0\}$. The rest of the details can be found in Section 5, specifically $n = 200$ and $k = 5$. The distance between the base density and the target density (1.85) provides a scale reference. The method is robust for the sampling distribution and for broad range of $w$ we reach essentially the same accuracy as when sampling from the target itself.

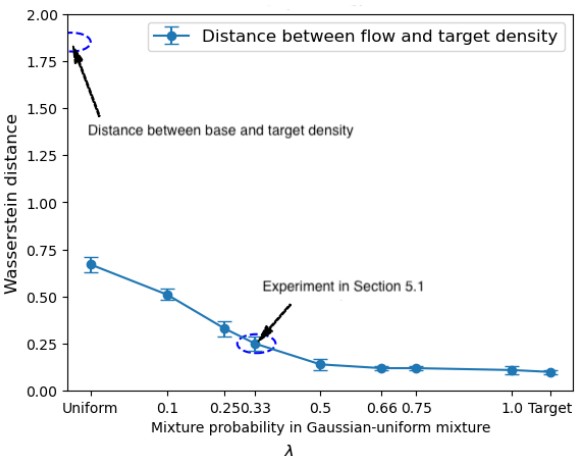

Table D.2: Wasserstein distances for varying $n$ (fixed $k = 5$) across different experiments

| $n$ | 25 | 50 | 100 | 1000 |
|---|---|---|---|---|
| Onemoon2D | 0.67 ($\pm 1.34$) | 0.18 ($\pm 0.04$) | 0.17 ($\pm 0.03$) | 0.23 ($\pm 0.02$) |
| Gaussian6D | 1.70 ($\pm 0.22$) | 1.50 ($\pm 0.19$) | 1.46 ($\pm 0.11$) | 1.26 ($\pm 0.04$) |
| $n$ | 50 | 500 | 2000 | 10000 |
| Funnel10D | 4.33 ($\pm 0.10$) | 3.96 ($\pm 0.05$) | 3.89 ($\pm 0.04$) | 3.92 ($\pm 0.04$) |
| Twogaussians10D | 2.69 ($\pm 0.31$) | 2.57 ($\pm 0.08$) | 2.61 ($\pm 0.05$) | 2.66 ($\pm 0.04$) |

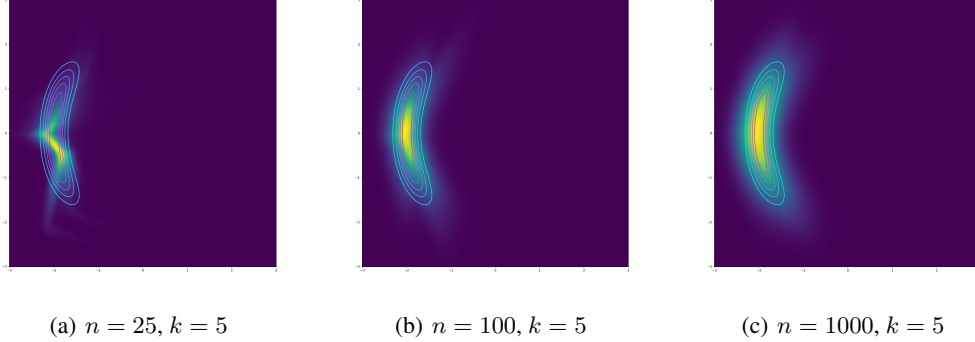

(a) $n = 25, k = 5$        (b) $n = 100, k = 5$        (c) $n = 1000, k = 5$

Figure D.4: The estimated belief densities in the Onemoon2D experiment of Table D.2 (contour: true density; heatmap: estimated flow). While the coverage of the estimated density with $n = 1000$ is better than with $n = 100$, there is more spread with $n = 1000$ than with $n = 100$, which explains the slight performance deterioration in Table D.2.

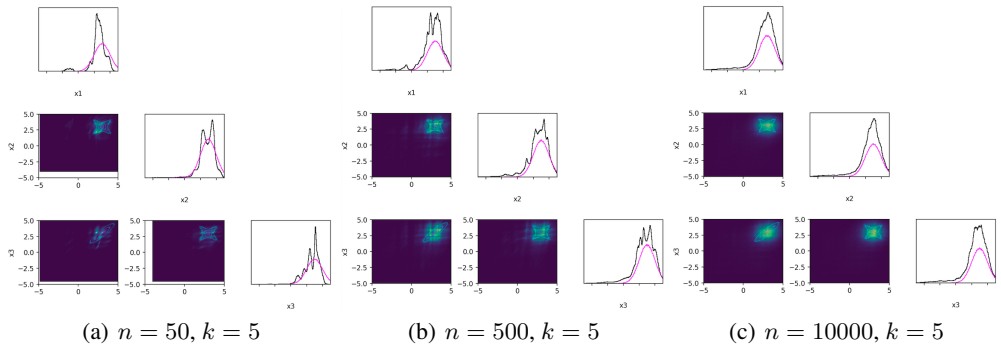

(a) $n = 50, k = 5$      (b) $n = 500, k = 5$      (c) $n = 10000, k = 5$

Figure D.5: Cross-plots of selected variables of the estimated flow in the Twogaussians10D experiment of Table D.2 (contour: true density; heatmap: estimated flow). While the coverage of the estimated density with $n = 10000$ is better than with $n = 500$, there is more spread with $n = 10000$ than with $n = 500$, which explains the slight performance deterioration in Table D.2.

## D.5 Effect of the cardinality of the choice set $k$

Finally, to complement the ablation studies for $k$ on synthetic settings in Section 5.3, we rerun the LLM expert elicitation experiment with $k = 2$. Figure D.6 shows that the LLM expert also works with $k = 2$. We replicated the original experiment conducted with $k = 5$ and report the estimates side-by-side, visually confirming we learn the essential characteristics of the distribution in both cases. The results are not identical and the case of $k = 5$ is likely more accurate (see e.g. the marginal distribution of the last feature), but there are no major qualitative differences between the two estimates.

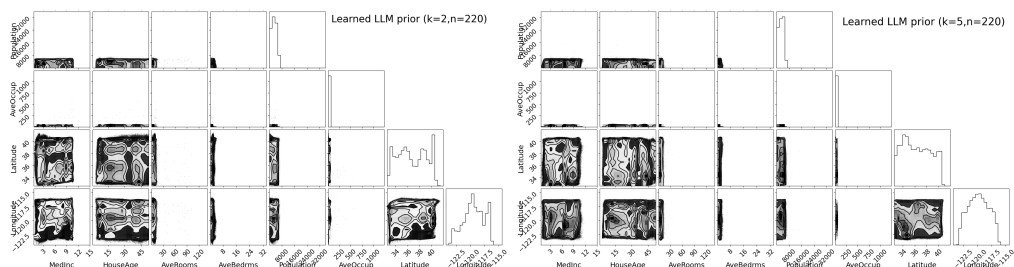

Figure D.6: The LLM expert elicitation experiment replicated for the setting of pairwise comparisons (left) and compared to the original setting of 5-wise rankings (right). The estimated flow remains qualitatively the same for the variables shown here (other variables omitted due to lack of space), and this holds true for the rest of the variables as well.

