# OpenReview forum: "Preferential Normalizing Flows"
_NeurIPS.cc/2024/Conference — NeurIPS 2024 poster_

### Official Review · Reviewer_xMnU · 2024-07-02

**Soundness:** 2
**Presentation:** 2
**Contribution:** 3
**Rating:** 5
**Confidence:** 2

**Summary:**

This paper focuses on the problem of eliciting a complex multivariate probability density from an expert. Existing works mainly use simple distributions. This paper proposes to model the belief density with a flow based model. To apply normalizing flows to this problem, we will need to address a few challenges. (1) We want to train flows with a small set of samples, which may result in collapsing and diverging probability mass. (2) The samples presented for the expert might be drawn from an independent and known distribution.

This paper proposes to use Bayesian inference to address the challenges of collapsing and diverging probability mass. That is, we are interested in representing the belief density $p_{}(x)$ with a flow model. We then optimize the parameters of $p_{*}(x)$ with MAP estimation. The likelihood of samples, i.e., $p(D|f)$ can be computed with Proposition 3.5 of the paper. The prior is approximated by the probability of the k-wise winners i.e., $p(D_{k}^{\succ})$.

The experiments on synthetic and real datasets show the effectiveness of the proposed method.

**Strengths:**

1. This paper applies normalizing flows to the problem of expert knowledge elicitation. I think this is a promising application of flow-based models. The idea is novel.

2. The proposed method is effective, and has theoretical guarantees.

**Weaknesses:**

Some definitions and formulas are not very clear to me.

1. In Proposition 2.1, it is not very clear what is the relationship between W and the limit $\lim_{\beta \rightarrow 0} p(\mathcal{D}) = 1$.

2. The symbol $\mathcal{D}$ is used in two places, i.e., Proposition 2.1 and Section 4.1

3. In Assumption 3, what is $Exp(s)$.

4. In Proposition 3.5, is $s()$ a function or a constant?

5. In Eq.4, Is the numerator $\exp{(sf(x)\lambda(x))}$?

6. The symbol $\mathbb{x}$ is used in several places. Sometimes, it represents a random variable, sometimes, it represents a sample.

7. In Eq. 6, is $f_i = f(x_i)$？

**Questions:**

1. I am not an expert of expert knowledge elicitation, so I may misunderstand something. Based on Eq. 1, the $f$ is a continuous function. Based on Eq. 6, $p(f) \propto \prod \exp{f_i}$. Therefore, why do we need normalizing flows to represent $f$. That is, it seems that we can use an arbitrary neural network to represent $f$.

2. What’s the meaning of Theorem 3.6. I may have missed something, but I did not see any discussion or analysis on this theorem.

3. The paper mentions that one challenge of this task is samples presented for the expert might be drawn from an independent and known distribution. How this challenge is addressed by the proposed method?

4. The experiments actually not very convincing, to improve the experiments,

    (1) Are there any existing methods that can be used as baselines?

    (2) Do we have any quantitative metrics to evaluate the proposed method?

**Limitations:**

In my opinion, the proposed method does not have potential negative societal impact.

---

> ### Author Rebuttal · Authors · 2024-08-07
>
> > In Proposition 2.1, it is not very clear what is the relationship between W and the limit \\(\lim_{\beta \rightarrow 0} p(\mathcal{D}) = 1\\).
>
> As the “noise level” \\(\beta\\) goes zero, there is no noise in RUM (i.e. \\(W=0\\) with high probability), which implies that the expert always chooses \\(\mathbf{x}\\) which maximizes their utility / belief density, that is \\(argmax_{\mathbf{x}} p_{\star}(\mathbf{x})\\). Only observations consistent with argmax have non-zero probability and hence the probability of the only possible data collection is one. Since argmax is invariant to monotonic transformations, one cannot identify the target belief density, or it is unidentifiable at least up to a monotonic transformation.
>
> > The symbol \\(\mathcal{D}\\) is used in two places, i.e., Proposition 2.1 and Section 4.1
>
> We modified the notation in Proposition 2.1 from \\(\mathcal{D}\\) to \\(\mathcal{D}_{\textrm{pref}}\\).
>
> > In Assumption 3, what is \\(Exp(s)\\)
>
> Exponential distribution with a rate parameter s>0. We will clarify this in the final version.
>
> > In Proposition 3.5, is \\(s()\\) a function or a constant?
>
> It is a constant, the rate parameter of the exponential distribution. The equation reads as \\(s\\) times \\(f(\mathbf{x}) - f(\mathbf{x}_j)\\), the parenthesis being for the difference and not indicating a function.
>
> > In Eq.4, Is the numerator \\(\exp{(sf(x)\lambda(x))}\\)?
>
> No, the equation in the paper is correct, i.e. the integrand in the numerator is \\(\exp{(sf(\mathbf{x}))\lambda(\mathbf{x})}\\). Note that \\(f(\mathbf{x}) := \log p_{\star}(\mathbf{x})\\), whereas $\lambda(x)$ is directly a density.
>
> > The symbol \\(\mathbb{x}\\) is used in several places. Sometimes, it represents a random variable, sometimes, it represents a sample.
>
> We used \\(X\\) for a random variable and \\(\mathbf{X}\\) for a set of samples (design matrix). To avoid confusion, we changed our notation so that the random variable is now denoted by \\(\mathbb{X}\\).
>
> > In Eq. 6, is \\(f_i = f(x_i)\\)?
>
> Yes. We will clarify this in the final version.
>
> > ...Based on Eq. 1, the \\(f\\) is a continuous function. Based on Eq. 6, \\(p(f) \propto \prod \exp{f_i}\\). Therefore, why do we need normalizing flows to represent \\(f\\)...
>
> The critical requirement is that the result has to be a density. More precisely, \\(\exp(f)\\) needs to normalised over the argument x, i.e. \\(\int \exp(f(\mathbf{x}))d\mathbf{x} = 1\\). Normalising flows provide a natural tool for representing densities, whereas for an arbitrary network we would need explicit normalization that is not computationally feasible.
>
> Further, Eq. (6) relates to the functional prior \\(p(f)\\), which defines a probability density over the *function values*.
>
> > What’s the meaning of Theorem 3.6. I may have missed something, but I did not see any discussion or analysis on this theorem.
>
> Theorem 3.6. provides the formula for the k-wise winner distribution in the limit of \\(k\\) (number of alternatives) approaches infinity. We use the limit result to construct the functional prior, which helps to solve the collapsing and diverging mass problem. Even though we eventually use only small \\(k\\), we can use the limit distribution as a prior, because we can temper the finite-k distribution to resemble the limit (Figure 2).
>
> > The paper mentions that one challenge of this task is samples presented for the expert might be drawn from an independent and known distribution. How this challenge is addressed by the proposed method?
>
> The whole method is designed specifically to address this challenge.
> If the samples were drawn from the target itself then standard flow learning would be sufficient as a solution, whereas learning the flow from preferential responses for samples drawn from any other distribution requires the full machinery we introduce. In other words, the challenge is addressed by the combination of the RUM model for preferential data and interpretation of the k-wise distribution as tilted version of the belief as described in Section 3, as well as the functional prior presented in Section 4. Further, the new ablation study (Figure R1) provides some insights on how the problem becomes more challenging when sampling distribution for the candidates is very different from the target density. Yet, the proposed method is able infer the target density, although the estimate is not accurate as it would be when the sampling distribution is closer to the target.
>
> > Are there any existing methods that can be used as baselines?
>
> Majority of prior elicitation works assume fixed prior family (e.g. a Gaussian) and would not be fair baselines; they can be made arbitrarily bad by making a poor choice of the distribution. Furthermore, there are no specific methods that learn from the preferential comparisons and hence conducting such a comparison would require deriving details for a new method anyway. While there are some methods that can estimate flexible densities (the GP-based methods we cited in Introduction), they all require completely different kind of input information and hence cannot be compared against in our setting. We learn from preferential comparisons, whereas e.g. Oakley \& O'Hagan (2007) learn from percentiles.
>
> Even though there are no natural baselines, we hope that the additional experiments relating to sensitivity of the method in terms of the key parameters (\\(k\\), \\(n\\), \\(\lambda(x)\\)) address in part the same request.
>
> > Do we have any quantitative metrics to evaluate the proposed method?
>
> Since the problem is a density estimation problem (from a preference data), a natural metric is the distance between the estimated density and the target density. The proposed method produces relatively low (Wasserstein and the MMTV) distances between the estimated flow density and the target density in the experiments.
>
> **References**
>
> Oakley, J. E., \& O'Hagan, A. (2007). Uncertainty in prior elicitations: a nonparametric approach. Biometrika

---

> > ### Comment · Reviewer_xMnU · 2024-08-10
> > **Response to authors**
> >
> > Thank you for your explanation. Your answers address my questions related to the theories of the proposed method. But I still have a concern on the experiments. Since there are no baselines for comparison, it is hard to tell how good the proposed method is, and what advantages the proposed method has. Especially I am not that familiar with expert knowledge elicitation.
> >
> > I notice that other reviewers are positive to this paper. It would be appreciated if other reviewers can explain to me the completeness and thoroughness of the experiments. Thank you.

---

> > > ### Comment · Reviewer_sR2g · 2024-08-11
> > >
> > > This is an important criticism that I will further consider. In my review, I listed a confidence as 3 because, while I'm confident reviewing the mathematics and model, I'm not familiar with comparable approaches capable of modeling an expert's belief density. I find the authors explanation as fair — specifically, that related approaches for modeling densities like these (e.g., GP-based) would require a different input and thus are not directly comparable. The formulation of the author's experiments with synthetic data are well designed, and the ablation studies help to dissect and analyze the model.

---

> > > > ### Comment · Reviewer_xMnU · 2024-08-12
> > > >
> > > > Thank you for your explanation. In my opinion, the paper theoretically and empirically demonstrates the proposed method, which it is very good. However, without any comparisons, a few questions are still not very clear.
> > > >
> > > > 1. What's the motivation of using flow-based models? That is, the method uses a flow-based model to represent $f(\mathbf{x})$. But is it necessary to use flow-based model? Especially, flow-based methods have some flaws, i.e., the collapsing and diverging probability mass mentioned in the paper. Comparing with traditional models, how much improvement we can get from the deep model?
> > > >
> > > > 2. When do we want to use the proposed method? That is, in what situations, we want to choose the flow-based method, and in what situations, we want to choose traditional methods.
> > > >
> > > > I am open to discussion. Thank you.

---

> > > > > ### Author Response · Authors · 2024-08-12
> > > > >
> > > > > We are happy to hear you are satisfied with the clarifications regarding the theoretical aspects.
> > > > >
> > > > > >What's the motivation of using flow-based models? That is, the method uses a flow-based model to represent \\(f(\mathbf{x})\\). But is it necessary to use flow-based model? Especially, flow-based methods have some flaws, i.e., the collapsing and diverging probability mass mentioned in the paper. Comparing with traditional models, how much improvement we can get from the deep model?
> > > > >
> > > > > To address this, we now made an explicit comparison against a baseline that does not use a flow to represent the density. We use the same preferential comparisons and optimize the same training objective, but instead of using a flow to represent  \\(exp(f(\mathbf{x}))\\) we directly assume the density is a *factorized normal distribution* parameterized by means and (log-transformed) standard deviations of all dimensions.  We remind that this exact method has not been presented in the previous literature, but was designed to validate the value of the flow representation. It retains the typical assumption of factorized priors made in the elicitation literature, but is learnt from the preferential comparisons as our method.
> > > > >
> > > > > Below we present an updated version of Table 1, including this new baseline and also introducing one new target distribution that is a mixture of two Gaussians. The flow is better in all cases with respect to all metrics, validating the value of learning a flexible distribution instead of fitting a parametric one. We visually validated the baseline works correctly, for instance fitting a Gaussian around the Onemoon2D density exactly as expected, and hence the difference is because of the flexibility of the distribution and not because of difficulties in learning the baseline.
> > > > >
> > > > > Note that for the Gaussian6D target the baselines makes the correct distributional assumption about the marginal distributions. However, the target has correlations between the dimensions and hence the proposed method still outperforms the baseline that is restricted to have diagonal covariance.
> > > > >
> > > > > **Table 1:** Accuracy of the density represented as a flow (*flow*) compared to a factorized normal distribution (*normal*), both learned from preferential data, in three metrics: log-likelihood, Wasserstein distance, and the mean marginal total variation (MMTV).
> > > > >
> > > > > | Dataset          | log-likelihood (↑)       | log-likelihood (↑)       | wasserstein (↓)        | wasserstein (↓)        | MMTV (↓)            | MMTV (↓)            |
> > > > > |------------------|--------------------------|--------------------------|------------------------|------------------------|---------------------|---------------------|
> > > > > |                  | **normal**                  | **flow**                  | **normal**               | **flow**               | **normal**            | **flow**            |
> > > > > | **Onemoon2D**    | -1.99 (± 0.09)            | -1.09 (± 0.12)            | 0.44 (± 0.04)          | 0.25 (± 0.04)          | 0.30 (± 0.02)       | 0.21 (± 0.02)       |
> > > > > | **Gaussian6D**   | -1.44 (± 0.09)           | -0.12 (± 0.02)            | 1.76 (± 0.08)          | 1.29 (± 0.05)          | 0.20 (± 0.02)       | 0.09 (± 0.01)       |
> > > > > | **Twogaussians10D** | -3.99 (± 0.07)        | -0.09 (± 0.01)            | 7.28 (± 0.14)          | 2.60 (± 0.06)          | 0.47 (± 0.01)       | 0.08 (± 0.00)       |
> > > > > | **Funnel10D**    | -2.23 (± 0.06)           | -0.09 (± 0.01)            | 5.13 (± 0.06)          | 3.92 (± 0.04)          |  0.27 (± 0.00)       | 0.18 (± 0.01)       |
> > > > > | **Abalone7D**    | -5.23 (± 0.07)            | -3.52 (± 0.09)            | 1.05 (± 0.01)          | 0.65 (± 0.01)          | -                   | -                   |
> > > > >
> > > > >
> > > > > > When do we want to use the proposed method? That is, in what situations, we want to choose the flow-based method, and in what situations, we want to choose traditional methods.
> > > > >
> > > > > The two key elements of the method are its ability to directly represent multivariate prior distributions without simplifying distributional assumptions and the requirement of only needing preferential comparisons, in contrast to standard methods that require experts to provide explicit information about e.g. quantiles.
> > > > >
> > > > > Consequently, it is the preferred method when we want easy human interaction (e.g. the expert is not trained in statistics and would find it difficult to relate to quantiles) and when we do not have reasonable grounds for assuming a specific distribution family. We believe the method will be particularly useful in applications where correlations between dimensions are expected, due to the very limited offering of multivariate parametric alternatives.
> > > > >
> > > > > Classical methods are likely to be more robust in scenarios where there is good theoretical basis for selecting a specific parametric prior and where the expert is qualified to directly provide information on quantiles or other statistical quantities. We will add a remark on this.

---

### Official Review · Reviewer_FQE4 · 2024-07-12

**Soundness:** 3
**Presentation:** 2
**Contribution:** 3
**Rating:** 8
**Confidence:** 3

**Summary:**

The paper proposes a method to learn so-called belief densities based on k-wise rankings or comparisons of alternatives. This belief density is learned by combining function-space Bayesian inference and normalizing flows, which allows for the learning of complex (multivariate) probability densities. The authors evaluate their method empirically by studying synthetic scenarios, a regression task and a “real” use-case by querying an LLM and evaluating how close the resulting flow is to the ground truth density by three metrics.

**Strengths:**

The paper addresses an interesting and very relevant problem, especially as human feedback has recently been used a lot in the training of LLMs. The paper has a strong theoretical contribution proposing a method how to learn the belief density using normalizing flows from rankings while preventing failure modes by using a functional prior.  The paper is well written and the authors clearly disentangles their contribution from the work in related works.

**Weaknesses:**

The experiments could have been more varied. At the moment all experiments use k = 5 and it would be interesting to see how the results vary with different values of k. Specifically, k = 2 is a very relevant real-life use case as this type of feedback is given a lot in e.g. chatbots. Similarly, it would be good to get some idea of how the flows converge to the ground truth density as a function of the number of preferential samples.

Minor points and typos:
Fig. 1 Can you make the numbers larger? They are hard to read even when zooming in and without zooming in it’s not clear that they are numbers
l. 78 “Diverging” -> “diverging”
l. 94 “Mollester” -> “Mosteller”
l. 201 “to Bayesian inference” -> “to use/do, etc. Bayesian inference”
l. 201 “given a preferential” -> “given preferential”
l. 405 citation ends abruptly
l. 458 “probality” -> “probability”

**Questions:**

Is assumption 3 satisfied in the LLM experiment? If not, how does this affect the results?

**Limitations:**

The paper mentions fair limitations in the discussion section.

---

> ### Author Rebuttal · Authors · 2024-08-07
>
> > ...At the moment all experiments use k = 5 and it would be interesting to see how the results vary with different values of k. Specifically, k = 2 is a very relevant real-life use case as this type of feedback is given a lot in e.g. chatbots. Similarly, it would be good to get some idea of how the flows converge to the ground truth density as a function of the number of preferential samples.
>
> Thank you for the excellent suggestion. We re-ran both Onemoon2D with \\(k=2,3,5,10\\) and LLM experiment with \\(k=2\\) to \\(k=5\\), as explained in the global response. The key result is that the method indeed works already with \\(k=2\\), though naturally not as well. We also studied the convergence as a function of \\(n\\), again describing the experiment and the results in the global response.
>
>
> > Minor points and typos:
>
> Thank you for pointing out these errors. We corrected the errors for the revised manuscript and increased the font size of Fig. 1.
>
> > Is assumption 3 satisfied in the LLM experiment? If not, how does this affect the results?
>
> The RUM model with this specific noise distribution serves as a theoretical model for the expert's choice, so it is highly unlikely that assumption 3 is satisfied in the LLM experiment. In fact, it most likely does not hold exactly for humans either. Hence this experiment shows that even under model misspecification that can be expected also in the real use-cases we can get sensible results. We hypothesize that a potential violation of assumption 3 can have similar effect as misspecification of the noise level, which can result in too high (or low) spread in the estimated flows (see e.g. 2nd col, 3rd row in Figure A.7).

---

> > ### Comment · Reviewer_FQE4 · 2024-08-09
> > **Response to authors**
> >
> > Thank you for the additional experiments and the clarification regarding assumption 3. I will increase my score to 8.

---

### Official Review · Reviewer_sR2g · 2024-07-13

**Soundness:** 3
**Presentation:** 4
**Contribution:** 4
**Rating:** 7
**Confidence:** 3

**Summary:**

The paper presents an approach for expressing an expert's belief density using a normalizing flow. Interestingly, the flow is trained solely on preferential questions (e.g., comparing and ranking) which follow a random utility model (RUM). The approach avoids several optimization issues that could occur when trying to model the more complex underlying data. The authors method relies on an approach that defines a functional prior over the preference data. The authors demonstrate their method on both simulated and real-world data, showing a very promising method for representing the belief density of expert opinions.

**Strengths:**

The authors overcome a unique challenge in modeling expert's beliefs with flows, namely that the true data distribution of experts beliefs $p^*$ cannot be directly sampled from. Instead, they sample from preference data $D$ and choose a specific function-space prior for the flow in order to model the likelihoods of the k-wise comparisons. This unique solution results in a flow that properly places mass on the winner points following the underlying decision model of the data. The experimental results confirm their hypothesis, showing much better results as a result of their prior.

Additionally, The use of LLMs as experts in their experiments is clever. The approach is reasonable and replicable, while also demonstrating and more end-to-end application that would be of interest to the broader community.

**Weaknesses:**

One primary advantage of normalizing flows is that they offer exact likelihood estimation. Because of the underlying data model for preference data (eq. 3), I suspect that the method here is actually optimizing a lower bound on the log likelihood. This may explain the optimization challenges and need for fixing the prior's precision $s=1$ described on lines 228-230. Specifically, their objective may be similar to the Max surjective flow described in Nielsen et. al. [2020, https://arxiv.org/abs/2007.02731]. In either case, the authors may want to clarify if their objective function is a lower bound or not, since adapting flows for non-bijective mappings is also a topic of interest to many.

**Questions:**

The Appendix explores various settings of the parameter $s$ when the true value of s varies, which shows that their choice to fix $s=1$ is reasonable. However, the authors almost mention using using $\lambda \propto 1$ as well to maintain tractability. The implications of this choice are unclear, did the authors run any validity checks to test the implications of their choice?

**Limitations:**

In practice collecting an training dataset of expert feedback is laborious, and often many experts are needed in order to collect enough data. The paper only considers the case of a single expert, with a flow model to that experts belief density. The authors may wish to comment on future work in this area — specifically, if their approach could be adapted to a multi-expert setting, if a single flow model is still sufficient or if a flow mixture model is preferred.

---

> ### Author Rebuttal · Authors · 2024-08-07
>
> >...Because of the underlying data model for preference data (eq. 3), I suspect that the method here is actually optimizing a lower bound on the log likelihood...
> Specifically, their objective may be similar to the Max surjective flow described in Nielsen et. al. [2020, https://arxiv.org/abs/2007.02731]...
>
> There are some similarities between the Max surjection transformation and the preferential likelihood given by (noiseless) RUM, which is essentially an argmax operator. However, the main difference is that the Max surjection transformation is a transformation that can be applied to any level of a composable transformation (denoted by \\(T\\) in the manuscript, by \\(f\\) in (Nielsen et. al., 2020)) with available likelihood contribution (Table 2, Nielsen et. al., 2020), while the RUM model conditions on the whole \\(T\\) and computes a (non-additive) likelihood contribution for the whole preferential flow.
>
> In SurVAE Flows, the likelihood contribution of a deterministic bijective flow is the likelihood of a datapoint \\(\mathbf{x}\\) given by the flow density \\(p\\) at that point, that is \\(p(\mathbf{x})\\) computed using the change of variable formula, which results in the Jacobian of \\(T^{-1}\\). In contrast, a preferential flow uses a probabilistic modelling approach by having an additional likelihood function \\(\mathcal{L}\\) which computes the likelihood contribution of (preference) data \\(\mathbf{x}\\) conditioned on the flow density \\(p\\), that is \\(\mathcal{L}(\mathbf{x} \mid p)\\). In a preferential flow, although the mapping \\(T\\) from a latent point \\(\mathbf{z}\\) to a datapoint \\(\mathbf{x}\\) is a deterministic bijective function, the likelihood contribution of preference data does not directly involve the Jacobian of \\(T^{-1}\\), unlike in SurVAE Flows (Algorithm 1, Nielsen et. al., 2020).
>
> Thus, the likelihood contribution is exact and the objective is not a lower bound. We extended the discussion section of the manuscript to discuss the relationship between SurVAE Flows and preferential flows.
>
> > The Appendix explores various settings of the parameter \\(s\\) when the true value of \\(s\\) varies, which shows that their choice to fix \\(s=1\\) is reasonable. However, the authors almost mention using using \\(\lambda \propto 1\\) as well to maintain tractability. The implications of this choice are unclear, did the authors run any validity checks to test the implications of their choice?
>
> We indeed use \\(\lambda \propto 1\\) for construction of the functional prior, but note that in all of the experiments we used as \\(\lambda(x)\\) a mixture of a bounded uniform and a Gaussian. That is, we already conducted the experiments under conditions where the choice does not hold.
>
> To further investigate the effect of the true sampling distribution, we now conducted an additional ablation study, where the true \\(\lambda(x)\\) is varied. See the global response for description of the experiment and summary of the results confirming that the method works for a range of choices.
>
> >...The paper only considers the case of a single expert, with a flow model to that experts belief density. The authors may wish to comment on future work in this area — specifically, if their approach could be adapted to a multi-expert setting, if a single flow model is still sufficient or if a flow mixture model is preferred.
>
> This is a good idea and we will extend the Discussion to cover also multi-expert settings.
>
> The most common methods for aggregating expertise of multiple experts are *behavioral aggregation* and *mathematical aggregation*. The former is based on experts discussing their opinions and making consensus judgments for which an aggregate distribution is fitted (O'Hagan, 2019). This is agnostic to the elicitation algorithm and our method with a single flow could readily be used to capture the consensus. For *mathematical aggregation* we need a *pooling rule* that combines the elicited densities into a single one (EFSA, 2014). We think that a mixture of flows, which is equivalent to a linear pooling over multiple elicited flow densities, with equal mixture weights, would be a good default option. Further, Bayesian pooling such as having a hierarchical model on the joint preferential data from multiple experts could be investigated.
>
> **References**
>
> EFSA, E. F. S. A. (2014). “Guidance on Expert Knowledge Elicitation in Food and Feed Safety Risk Assessment.” EFSA Journal, 12(6): 3734. 2, 16\\
> O’Hagan, A. (2019). Expert knowledge elicitation: subjective but scientific. The American Statistician, 73(sup1), 69-81.

---

> > ### Comment · Reviewer_sR2g · 2024-08-11
> >
> > Thank you for clarifying these points, the additional ablation study, and further discussion. The extended discussion connecting preferential flows to other flows will be a nice contribution for the community. Additionally, confirming that the preferential flows you present have exact likelihood contributions is a strength and may lead to additional future applications.

---

### Author Rebuttal · Authors · 2024-08-07

We are happy to see the reviewers both understood the paper well and perceived it positively. We thank the reviewers for the detailed constructive comments, and provide responses to specific comments and questions for each reviewer separately.

The rebuttal is accompanied by a pdf that reports results of new empirical experiments addressing comments from reviewers sR2g and FQE4, validating the sensitivity of the results in terms of the cardinality of the choice set \\(k\\), the number of comparisons \\(n\\), and the choice of the distribution \\(\lambda(x)\\) that the candidates are sampled from.

We report results for three new experiments:

1. Figure R1 studies the effect of \\(\lambda(x)\\), the unknown distribution from which the candidates to be compared are sampled from, complementing the experiment reported in Section 5.1 and confirming the method is robust for the choice. In the original experiment the candidates were sampled from a mixture distribution of uniform and Gaussian distribution centered on the mean of the target, with the mixture probability \\(w=1/3\\) for the Gaussian. Figure R1 reports the accuracy as a function of the choice of \\(w\\) for one of the data sets (Onemoon2D), so that \\(\lambda(x)\\) goes from uniform to a Gaussian,
and includes also an additional reference point where \\(\lambda(x)\\) equals the target. For all \\(w>0.5\\) we reach effectively the same accuracy as when sampling from the target itself, and even for the hardest case of uniform \\(\lambda(x)\\) (with \\(w=0\\)) the distance is significantly smaller than the reference scale comparing the base distribution with the target.

2. Tables R1 and R2, as well as Figure R2, report results of an experiment studying the effect of \\(k\\) and \\(n\\). Again the results are reported for only one of the data sets (Onemoon2D), but we will add similar results for all of the data sets in Appendix.
For fixed \\(n\\), we see that the accuracy naturally improves as a function of \\(k\\) (Table R1). The original manuscript used \\(k=5\\), but the new result reveals that we can learn the target already with \\(k=2\\) that is most convenient for a user, but naturally with somewhat lower accuracy. For fixed \\(k\\), increasing \\(n\\) generally improves the accuracy and already fairly small \\(n\\) is sufficient for learning a good estimate (Table R2). For very large \\(n\\) the accuracy can slightly deteriorate. We believe this is because of prior misspecification that encourages overestimation of the variation due to the fact that \\(k\\) is finite but in the prior it is assumed to be infinite. Figure R2 confirms that for \\(n=1000\\) the shape of the estimate is extremely close and the slightly worse Wasserstein distance is due to overestimating the width. For the final version, we extend the second paragraph of the Discussion to elaborate this aspect more.

3. Finally, Figure R3 shows that the LLM expert also works with \\(k=2\\). We replicated the original experiment conducted with \\(k=5\\) and report the estimates side-by-side, visually confirming we learn the essential characteristics of the distribution in both cases. The results are not identical and the case of \\(k=5\\) is likely more accurate (see e.g. the marginal distribution of the last feature), but there are no major qualitative differences between the two estimates.

---

### Decision · Program_Chairs · 2024-09-25

**Decision:**

Accept (poster)

**Comment:**

This work proposes a method of learning an expert’s belief density in which the density is parameterized as a normalizing flow.  The main interesting elements are that the expert information is provided solely through preferential data (e.g., k-wise winner of a set), and that the authors devise a functional prior for the normalizing flow based on the choice density, or the distribution of elements selected to be present in the preferential sets.  They argue the reasonableness of this choice and show that empirically it helps them avoid density collapse during learning.  Reviewers felt that the experiments were somewhat limited, with a lack of alternatives or baselines to give a sense of the quality of the method.  The application of eliciting expert belief from LLMs is neat.

There are also a number of typos in the draft that should be fixed if accepted.